# Confinement anisotropy drives polar organization of two DNA molecules interacting in a nanoscale cavity

Zezhou Liu [1✉], Xavier Capaldi[1], Lili Zeng [1], Yuning Zhang [1,2], Rodrigo Reyes-Lamothe [3] & Walter Reisner[1✉]

There is growing appreciation for the role phase transition based phenomena play in biological systems. In particular, self-avoiding polymer chains are predicted to undergo a unique confinement dependent demixing transition as the anisotropy of the confined space is increased. This phenomenon may be relevant for understanding how interactions between multiple dsDNA molecules can induce self-organized structure in prokaryotes. While recent in vivo experiments and Monte Carlo simulations have delivered essential insights into this phenomenon and its relation to bacteria, there are fundamental questions remaining concerning how segregated polymer states arise, the role of confinement anisotropy and the nature of the dynamics in the segregated states. To address these questions, we introduce an artificial nanofluidic model to quantify the interactions of multiple dsDNA molecules in cavities with controlled anisotropy. We find that two dsDNA molecules of equal size confined in an elliptical cavity will spontaneously demix and orient along the cavity poles as cavity eccentricity is increased; the two chains will then swap pole positions with a frequency that decreases with increasing cavity eccentricity. In addition, we explore a system consisting of a large dsDNA molecule and a plasmid molecule. We find that the plasmid is excluded from the larger molecule and will exhibit a preference for the ellipse poles, giving rise to a non-uniform spatial distribution in the cavity that may help explain the non-uniform plasmid distribution observed during in vivo imaging of high-copy number plasmids in bacteria.

---

[1] Department of Physics, McGill University, 3600 rue université, Montréal, QC H3A 2T8, Canada. [2] BGI-Shenzhen, Shenzhen 518083, China. [3] Department of Biology, McGill University, 33649 Sir William Osler, Montreal, QC H3G 0B1, Canada. ✉email: zezhou.liu@mail.mcgill.ca; reisner@physics.mcgill.ca

Biological systems exploit phase transition physics to ensure their proper organization and function[1]. Liquid-liquid phase transitions are now believed to account for the formation of membrane-less organelles, such as P granules[2]; 2D phase separations between liquid-disordered and liquid-ordered lipid phases[3] may give rise to the phenomenon of lipid microdomains (lipid rafts)[4]. In these classic examples, the phase separation is driven by the collective weak interaction of many small molecules (e.g., proteins, lipids).

Phase separation can also be induced by the interaction of larger but less numerous dsDNA polymer molecules[5]. Counterintuitively, whereas entropy maximization favors mixing of small particles in the absence of attractive interactions, long polymer chains are predicted to demix, due to the higher excluded volume and thus lower entropy, of the non-mixed conformations[6]. A typical prokaryotic cell contains multiple large and freely interacting DNA molecules, such as primary/secondary chromosomes[7] and plasmids. As prokaryotes lack a separate nuclear compartment, these multiple dsDNA molecules are free to interact within the cell volume. Entropy-driven demixing of dsDNA molecules can thus affect internal prokaryotic organization and function. Jun et al. famously suggested that entropic polymer demixing could provide the driving force behind bulk chromosomal segregation in dividing bacteria[5,6]. More recently, entropic polymer demixing has attracted attention as a possible mechanism to explain the non-uniform distribution of plasmids observed in live-cell imaging of *E. coli*[8–11], including a tendency for plasmids to localize at the poles[8] and in a ring at the periphery of the bacterial chromosome[9]. This may in turn have implications for the partitioning of high-copy number plasmids upon cell division[10].

A remarkable property of entropically driven polymer demixing is that the predicted mixing-demixing phase-space depends on the anisotropy of the imposed confinement; demixing is believed to be greatly enhanced in tube-like structures (e.g., nanochannels and rod-like bacteria)[6,12]. While this phenomenon is predicted by classic scaling theories[6] and Monte Carlo simulation[12–15], key questions remain regarding exactly how multiple polymer states are internally organized and fluctuate dynamically in confined volumes of varying anisotropy. How, in particular, does polar organization develop in a system of two confined polymers as the rotational symmetry of an initially isotropic confined volume is broken? What dynamic features emerge when anisotropy is introduced? Can polar organization develop spontaneously in a confined anisotropic system consisting of one large polymer and additional smaller polymer molecules (e.g., plasmids)? These fundamental polymer physics questions may have relevance for how shape anisotropy influences the organization of demixed polymer states in the corresponding bacterial systems (e.g., rod-like versus spherical versus box-shape bacteria).

Using in vivo methods to probe these questions is challenging, due to the immense complexity of the biological systems—involving many overlapping molecular processes—and the inability in vivo to independently control physically essential system parameters without drastically altering cellular phenotype and functionality. In addition, as physical and active mechanisms can interact in complex ways, teasing out their distinct roles is not straightforward. For example, biological systems may exploit polymer-driven demixing in certain contexts (e.g., entropy as a driver of chromosomal segregation), working in concert with active systems[16], while in others the same physical effect may be biologically undesirable (e.g., polymer demixing can expel large plasmids from the nucleoid), so that additional active mechanisms are needed (e.g., a special partitioning system for large plasmids)[11]. Lastly, focusing only on specific in vivo systems may obscure understanding of how the system behaves physically over a larger parameter space (i.e., a parameter space defined in terms of gross biophysical parameters like cell size, degree of anisotropy in the cell geometry, number of chromosomes/plasmids, sizes of chromosomes/plasmids and degree of crowding). Specific in vivo systems occupy only narrowly defined regions of this space. However, exploring the physical behavior over much larger portions of the space, even parts of the space that do not contain viable organisms, is essential to probe the underlying physics and can place existing in vivo systems in a larger context, for example shedding light on differences between species that occupy different points in parameter space[6], or physical constraints critical for cellular viability.

In this communication we develop a drastically simplified model system, containing two dsDNA molecules interacting in an elliptical nanoscale compartment (Fig. 1), to serve as a minimal model to explain how polymer-polymer interaction in anisotropic confinement can give rise to states with polar organization. Note that our choice of an elliptical geometry is designed to emphasize behavior that arises purely from confinement anisotropy, rather than features that might arise from geometries specifically mimicking a given biological system. Firstly, by confining two differentially stained dsDNA molecules of equal size in elliptical compartments of varying eccentricity (Fig. 1a), we demonstrate that increasing compartment anisotropy leads to a symmetry-breaking phenomenon whereby the chains segregate to either side of the elliptical boxes. This polar organization of the two chains at the ellipse poles can be understood as an orientational configuration transition that can be quantified by an order parameter analogous to that used for a liquid crystal isotropic to nematic transition. Secondly, by confining a larger dsDNA molecule and a

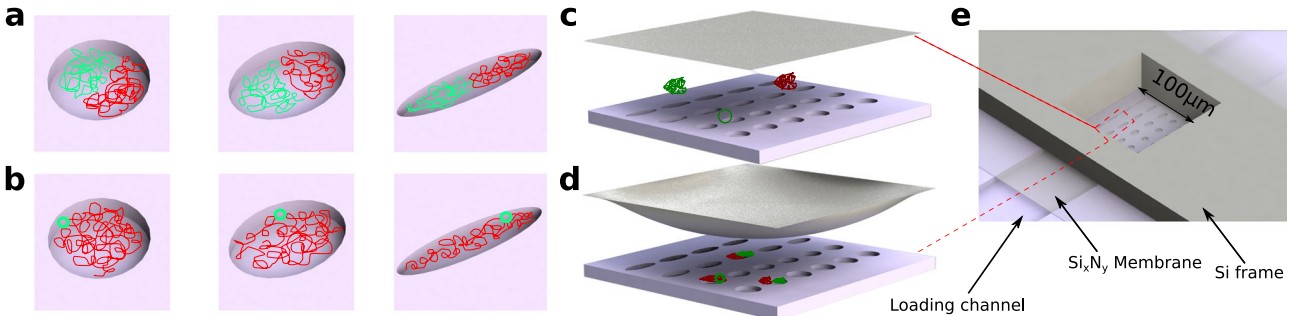

**Fig. 1 Schematic of the experimental concept and setup. a** Schematic of two differentially stained equal-sized dsDNA molecules confined in elliptical compartments of varying eccentricity. **b** Schematic of a larger dsDNA molecule and a plasmid molecule confined in the same structures. **c**, **d** Molecular confinement is induced mechanically by using pneumatic pressure to depress a thin membrane lid. Depression of the lid traps the molecules in nanoscale cavities embedded in the floor of a nanoslit flow-cell. **e** Zoomed-out view of device: the cavities are defined in a nanoslit that is interfaced to the flexible membrane lid at a central window etched through a supporting silicon frame.

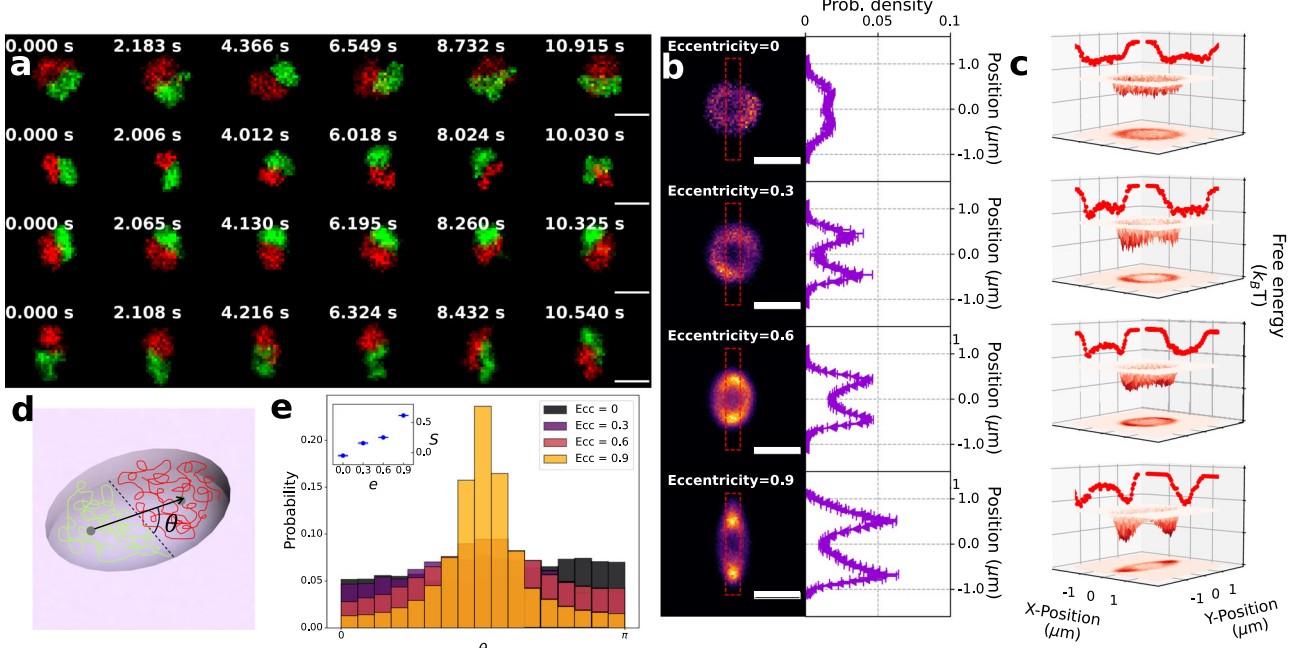

**Fig. 2 Experimental results and equilibrium analysis for two cavity confined DNA chains of equal size. a** Montage of fluorescence images of two $\lambda$-DNA molecules confined in elliptical cavities with varying eccentricity (the eccentricity values, ranging top to bottom, are 0.0, 0.3, 0.6 and 0.9). The scale bars are 2 µm and the time lapse between consecutive images is about 2 s. **b** Probability of finding a chain center at certain position within the cavity. The cross-section of the probablity density distribution, labeled with red dashed box, is shown next to the heatmap. The error bars denote the standard deviation of the probability density. **c** Free energy landscape within the cavity. The red lines indicate the projection of the landscape along the corresponding direction. **d** A cartoon giving the definition of the separation vector and $\theta$. **e** The resultant $\theta$-distribution for cavities of varying eccentricity. The inset shows the extracted order parameter. The error bars denote the standard error of the mean.

small plasmid in the aforementioned structures (Fig. 1b), we show that entropy-driven demixing can give rise to a ring-like and polar distribution of the plasmids, a phenomenon we find enhanced in the presence of molecular crowding. In particular, we find that the plasmid's polar distribution is driven by the symmetry mismatch between polymer-polymer exclusion and wall-polymer interaction.

## Results

### Polar organization of two DNA chains confined in an anisotropic cavity.

Our experimental system is based on a nanofluidic device consisting of an array of elliptical cavities embedded in a nanoslit (Fig. 1c–e). The cavities have eccentricities $e$ ranging from 0 to 0.9 (Fig. 1a, b) and are designed so that their volume is held constant as the eccentricity is increased. The nanoslit is bonded to a flexible lid that can be deflected downwards via pneumatic pressure, trapping molecular species in the cavities (Fig. 1c, d), a principle now exploited in a number of single-molecule confinement studies[17–22]. The cavity devices are etched 200 nm deep and have a maximum diameter that ranges from 2 µm (for $e = 0$) to 3 µm (for $e = 0.9$).

In our first experiment, we introduce $\lambda$-DNA into the device; $\lambda$-DNA has a gyration radius of 0.7 µm, sufficiently large so that when two $\lambda$-DNA chains are trapped in a single cavity their lateral conformation and organization will be influenced by the confinement. The $\lambda$-DNA consists of a mixture of molecules stained with two different dyes (YOYO-1 and YOYO-3). Once driven beneath the flexible membrane via pressure-actuated flow, the molecules are isolated in the cavities by depressing the lid, a procedure that is repeated until two differentially stained molecules are trapped, enabling independent monitoring of their conformation. Figure 2a shows a montage of fluorescence

micrographs of the cavity confined chains. For a cavity with $e = 0$, the molecules displace each other towards the cavity edges, forming an opposed pair that undergoes brownian rotation about the cavity center. This effect, which follows closely the behavior observed in a square cavity[21], is driven by volume exclusion between two-chains, which leaves the center of the cavity unfavorable to the two chains. When cavity eccentricity is introduced, the rotational symmetry is broken, and the molecules spend more time at the ellipse poles. At high eccentricity, the two chains adopt a strongly polar organization, with the chains stochastically swapping poles after a certain dwell-time.

To quantify the symmetry breaking of the $\lambda$-DNA spatial organization, we extract the chain positions $\mathbf{r}_1$ and $\mathbf{r}_2$ by computing the fluorescence center-of-mass for each chain. Histograms of the combined $\mathbf{r}_1$ and $\mathbf{r}_2$ values yield the probability $P_{\mathrm{CM}}$ of finding a chain center at some position within the cavity (Fig. 2b). The corresponding free-energy landscape $F_{\mathrm{CM}} = -k_B T \log P_{\mathrm{CM}}$ is also shown (Fig. 2c). Figure 2b, c clearly indicate the breaking of rotational symmetry as the cavity anisotropy is increased. For a cavity with $e = 0$, $P_{\mathrm{CM}}$ has a donut shape, consistent with Brownian rotation of chains symmetrically displaced from the cavity center. For $e = 0.3, 0.6$, $P_{\mathrm{CM}}$ appears as an elliptical donut ($e = 0.3, 0.6$). For $e = 0.9$, $P_{\mathrm{CM}}$ is peaked at the cavity poles, indicating fully polar segregation. This behavior reflects the underlying evolution of the two chain free-energy landscape from a ring to a double-well shape. For eccentricity values below $e = 0.6$, the free energy minima lies near the cavity rim and circuits the ellipse. At $e = 0.6$ distinct free energy wells form at the ellipse poles, indicating a fully polarized state.

The separation vector $\mathbf{r} = \mathbf{r}_1 - \mathbf{r}_2$ serves as an additional measure of symmetry breaking, tracking the self-alignment of the two-chain system along the cavity long-axis. Let $\theta$ correspond to the angle between $\mathbf{r}$ and the ellipse semi-minor axis (Fig. 2d). The

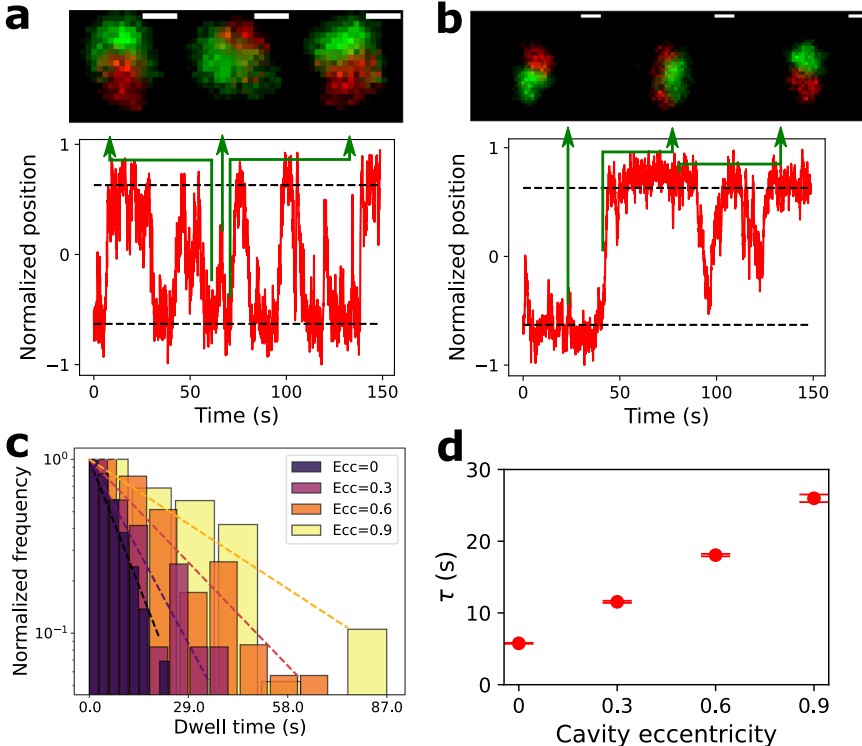

**Fig. 3 Experimental results and dwell time analysis for two cavity confined DNA chains of equal size. a, b** Time series of separation vector projected along the semi-major axis (red-curve) for $e = 0.6$ and $e = 0.9$ respectively. The projections are normalized to half the cavity extent, so localization of the configuration in a polar proximal free energy well corresponds to a value close to ±1. The black dashed line shows the baseline of two states. The green solid lines indicate the time corresponding to the adjoining image. In **a**, the two chains attempt to flip but fail to do so, forming a spike in the vector projection. In **b**, the two chains flip successfully, reversing the sign of the vector projection. The scale bars are 1 μm. **c** Histogram of the dwell time for four eccentricities; overlaid dashed lines correspond to exponential fits. **d** Mean dwell time verses cavity eccentricity, with the mean dwell time extracted from exponential fitting to the corresponding cumulative distributions. The error bars denote the fitting covariance.

distribution of $\theta$ (Fig. 2e), as eccentricity is increased, moves from a flat distribution indicating no alignment to a distribution peaked at $\frac{\pi}{2}$, indicating that the two-chain system aligns along the semi major axis. The peaking of the angular distribution leads to a corresponding increase in the order parameter $S = 2\langle \cos^2(\theta) - \frac{1}{2} \rangle$.

Molecules trapped in the polar proximal free energy wells can stochastically swap their position due to thermally assisted escape across the free energy barrier (Fig. 2c). We measure the stochastic pole swapping by monitoring the projection of **r** along the semi-major axis. This quantity changes sign when a swap occurs. Figure 3a, b give an example of the time-series of the projected separation vector normalized to half the cavity extent (for $e = 0.6$ and $e = 0.9$). The time series indicate a two-state profile. When two chains attempt to swap positions but fail, a sharp peak (or dip) in the projected value will arise and the separation vector will revert to its original value (e.g., see Fig. 3a, green solid line). A successful attempt forms a raising (or falling) edge accompanied by a flip in the separation vector polarity (e.g., see Fig. 3b, green solid line). With increasing eccentricity, swapping events become less frequent.

The system dwell-time $\Delta t$ in a given polar proximal state is exponentially distributed (i.e., $P(\Delta t) \sim e^{-\frac{\Delta t}{\tau}}$), as expected for a system with a constant escape rate[23] (Fig. 3c). We observe that the average dwell-time $\tau$, extracted from the exponential fits to the cumulative probability distribution (see Supplementary Note 1), is monotonically increasing with the cavity eccentricity (Fig. 3d). The increased average dwell-time is consistent with the increasing free energy barrier between the two pole proximal free energy wells (Fig. 2c). As illustrated in Kramers' expression[24],

$\tau \sim e^{\frac{\Delta F}{k_B T}}$ with $\Delta F$ is the free-energy barrier between two states. Qualitatively, note that increasing cavity eccentricity limits space in the cavity waist, yielding a higher free-energy barrier for the two chains to squeeze past each other in pole-reversal (Fig. 3b).

**Entropy-driven plasmid segregation.** Next, we explore the interaction between a larger, linear DNA molecule ($T_4$-DNA, 166 kbp) and a plasmid vector (pBR322, 4361bp) confined in an elliptical cavity (Fig. 4). The molecules are differentially stained, as before. Figure 4a gives a montage of fluorescence micrographs of the interacting molecules. For cavities with low eccentricity, the plasmid tends to reside at the cavity periphery, diffusing in a narrow band between the $T_4$-DNA and the cavity side walls. As the eccentricity increases above 0.9, the plasmid shows a preference for the cavity pole, yet undergoes stochastic switching between the poles by sliding between the $T_4$-DNA and the cavity side-wall. These qualitative observations on plasmid localization are confirmed via histograms of the plasmid position in the presence of the $T_4$-DNA for each eccentricity (Fig. 4b). Evidently, while the plasmid can penetrate the $T_4$-DNA, exclusion is sufficiently strong to ensure that the plasmid is most probably located on a ring circumventing the cavity periphery. For cavity eccentricity greater than 0.9, peaks of plasmid localization probability at the poles becomes evident, and there is a suppression of localization probability in the cavity mid-section. The $T_4$-DNA is centered in the cavity with remarkable precision, with a standard error of the mean of the center-of-mass position less than 1% of the cavity width. Self-centering of DNA in live *E. coli* is also reported by Wu et al.[25] Our experiment suggests that the self-

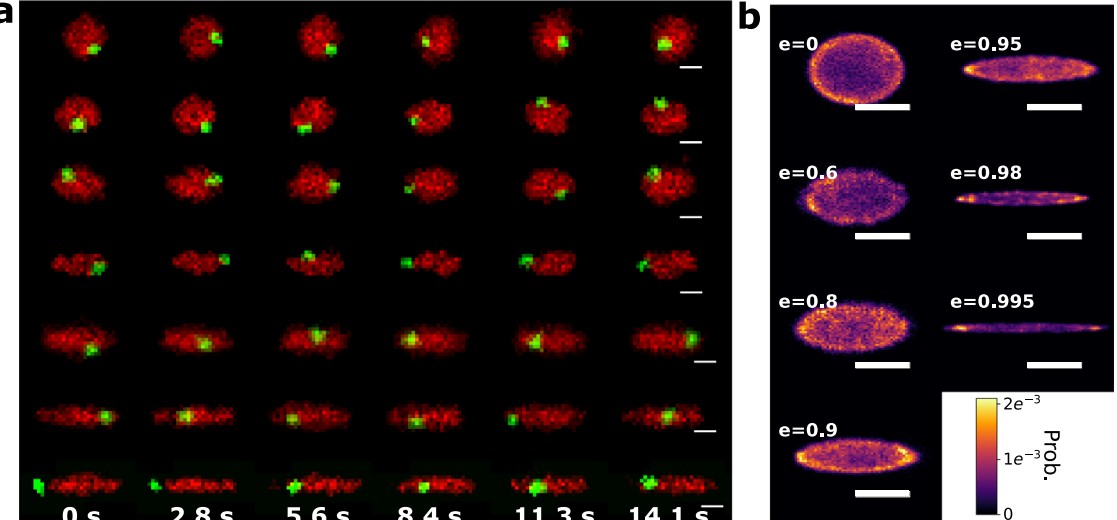

**Fig. 4 Experimental results and equilibrium analysis for a cavity confined plasmid in presence of T₄-DNA. a** Montage of fluorescence images of cavity confined T₄-DNA and plasmid DNA for cavities of varying eccentricity. The red color indicates T₄-DNA while the green indicates plasmid DNA. The scale bars are 2 μm. **b** Histogram of plasmid center-position while confined with T₄-DNA, yielding an estimate of the probability density function for plasmid position in the cavity. The scale bars from top to bottom correspond to 1 μm, 1 μm, 1 μm, 1.2 μm, 1.2 μm, 1.6 μm and 1.6 μm.

centering of DNA chains can be achieved by conformational entropy regulated by the confinement geometry.

**A model for plasmid segregation.** We hypothesize that the observed plasmid probability density arises from the competition of two effects: (1) plasmid exclusion from regions of high T₄-DNA concentration and (2) a repulsive interaction of the plasmid with the cavity boundary. As the gyration radius of the T₄-DNA (~1.5 μm) is comparable to the cavity size, we treat the T₄-DNA as a semi-dilute polymer solution with a concentration profile given by the density function $\rho_{T_4}(\mathbf{r})$. The quantity $\mathbf{r}$ corresponds to a 2D position vector in the cavity. We evaluate $\rho_{T_4}(\mathbf{r})$ using a mean-field approach with ground-state dominance[26] appropriate for the slit-confinement[27]. In this approach, the concentration profile is determined by solution of a non-linear Schrödinger equation with $\rho_{T_4} = 0$ imposed at the cavity boundaries[27]. The corresponding interaction potential between the plasmid and the T₄-DNA at position $\mathbf{r}$ is then proportional to $\rho_{T4}(\mathbf{r})$: $U_{T_4}(\mathbf{r}) = a\rho_{T_4}(\mathbf{r})$, with $a$ being a proportionality constant related to the strength of exclusion between the plasmid and the T₄-DNA.

We argue that the plasmid interacts with each patch of arc-length $ds$ along the cavity boundary via a potential $u_{\text{wall}}(\mathbf{r}_s - \mathbf{r})$, a function of the distance from the plasmid center position ($\mathbf{r}$) to the position of the particular boundary segment $ds$ at arclength $s(\mathbf{r}_s)$. We choose an exponential form for $u$: $u_{\text{wall}}(\mathbf{r}_s - \mathbf{r}) = b\exp\left(-\frac{|\mathbf{r}_s - \mathbf{r}|}{r_b}\right)$, simply reflecting an interaction that decays over a length scale $r_b$ (on order of magnitude of the distance the plasmid maintains from the cavity boundary). The quantity $b$ characterizes the strength of the wall-depletion effect. In order to obtain the total boundary-interaction potential $U_{\text{wall}}(\mathbf{r})$, we integrate the contributions from each patch along the cavity boundary:

$$U_{\text{wall}}(\mathbf{r}) = \oint u_{\text{wall}}(\mathbf{r}_s(s) - \mathbf{r})ds \qquad (1)$$

The plasmid explores the potential landscape stochastically via Brownian diffusion (Fig. 4a) with a particle position distribution $P(\mathbf{r})$ following the Boltzmann distribution $P(\mathbf{r}) \sim \exp(-U_p(\mathbf{r})/k_BT)$[24,28]; the potential experienced by the plasmid $U_p(\mathbf{r}) =$

$U_{T_4}(\mathbf{r}) + U_{wall}(\mathbf{r})$. Numerical solution of the concentration profile $\rho_{T_4}(\mathbf{r})$ determines $U_{T_4}(\mathbf{r})$; this is performed using an open-source finite element PDE solver FreeFEM (see Supplementary Note 2)[29]. We fit the values of the parameters $a$, $b$ and $r_b$ by finding the values that maximize the cosine similarity[30] between the experimental plasmid position distribution and the modeled plasmid position $P(\mathbf{r})$.

The model plasmid probability density is shown in Fig. 5a, binned down to the same spatial resolution as the experimental results (~50 nm). We observe that the model qualitatively matches our experimental results, with the plasmid circumferential ring-shaped distribution and pole preference evident. For a more quantitative comparison with experiment, we compare the cross-section of the experimental and modeled plasmid probability distribution along the major- and minor- axis of the ellipse (Fig. 5b, c gives the comparison for a cavity with $e = 0.9$. The cross-sections for additional cases are shown in Supplementary Note 3). Note that the positions of peak plasmid probability along the major axis, yielding the degree of polar segregation, agree well with the fitted model. The model also correctly describes the maximum concentration along the minor axis, which gives the position of the concentration ring. To explain intuitively why the plasmid segregates to the poles, we draw a portrait of the net potential experienced by the plasmid, plotting $U_{T_4}$ and $U_{wall}$ along the major axis (Fig. 5d, e). Note that the competition between the two potentials yields a segregation zone between the cavity center and the wall boundary. However, increasing the cavity eccentricity breaks the rotational symmetry of the potential landscape, so that the potential valley becomes deeper along the segregation zones that parallels the major axis. Also, the width of the valley, which can be observed qualitatively from Fig. 5d, e, is wider along the major-axis. Note that, for $e > 0.9$, the agreement between the model and measured plasmid position breaks down; this necessarily arises as our modeled T₄-DNA concentration distribution is no longer accurate[27] when the cavity becomes so elongated as to become tube-like and DNA semiflexibility plays a significant role. In the future, this problem may be addressed by more accurate modeling of the confined T₄-DNA concentration profile using self-consistent mean field approaches that can incorporate DNA semiflexibility[31,32].

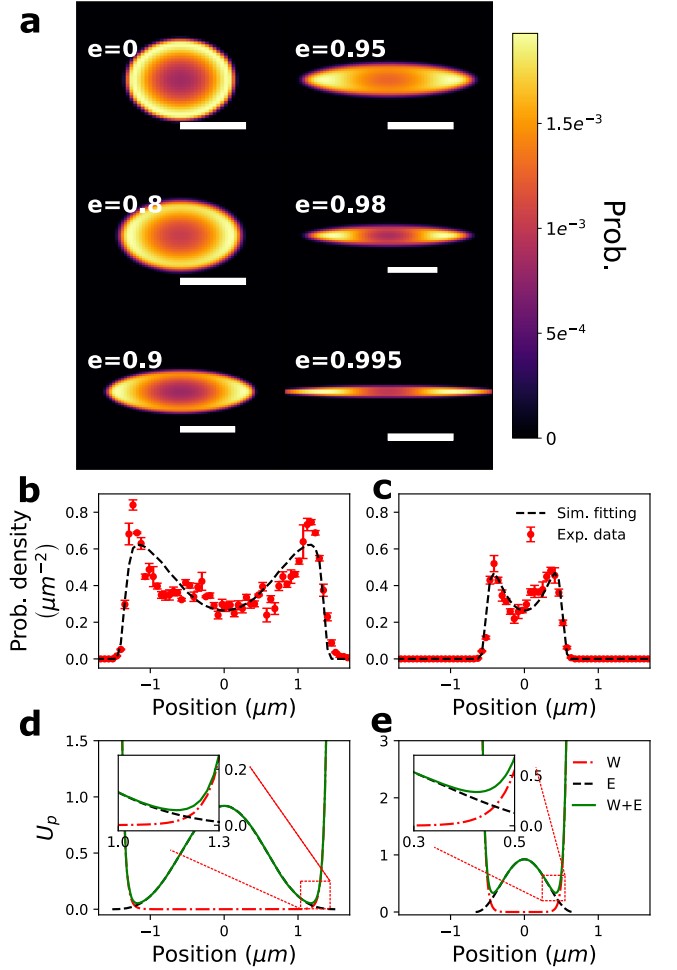

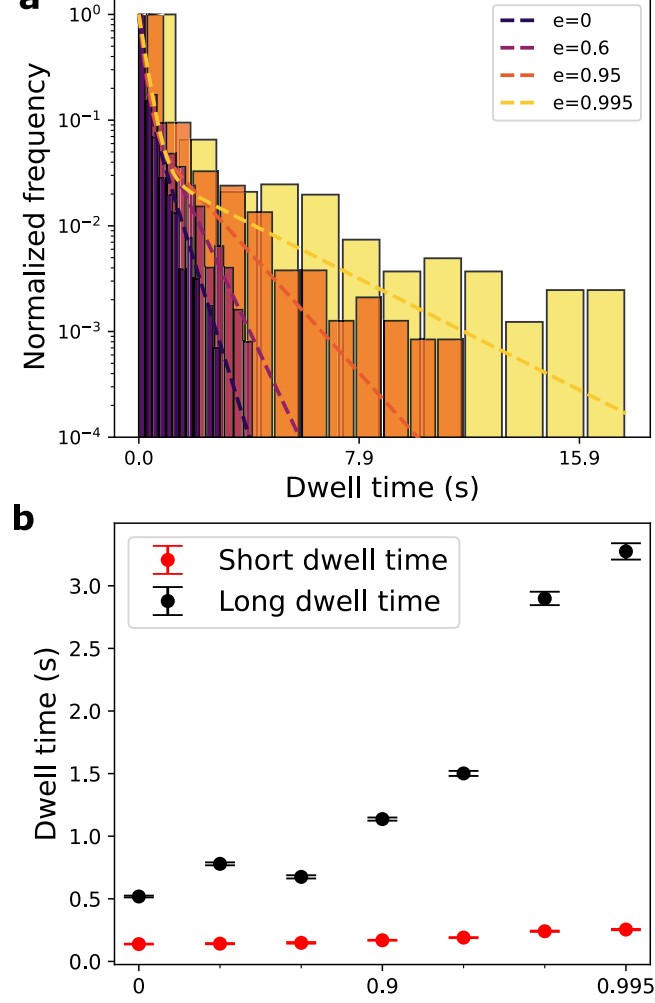

**Fig. 5 Modeling the distribution of plasmid position and comparison with experimental data. a** Fitted position distribution of the confined plasmid in cavities containing a single $T_4$-DNA molecule with eccentricity ranging from $e = 0$ to $e = 0.995$. The scale bars are 1 μm, 1 μm, 1.2 μm, 1.2 μm, 1.6 μm and 1.6 μm respectively for cavities with eccentricities ranging from $e = 0$ to $e = 0.995$. Plasmid probability density along the major (**b**) and minor (**c**) axis for $e = 0.9$. Experimental data are shown as red points, with error bars corresponding to the standard error of the mean of the binned counts ($n = 3$ bins for each point). Black dashed lines indicate the resulting fitted model plasmid probability density. Cross-sectional slices of the predicted potential along the major (**d**) and minor (**e**) axis for $e = 0.9$. The red dot-dashed line indicates the wall-potential; the black dashed line indicates the exclusion potential arising from the $T_4$-DNA; the green solid line indicates the superposition of both potentials. Note that a potential well forms at the overlap region between the repulsive wall-potential and the self-exclusion potential, with the insets giving the detailed behavior of the potential in the well vicinity.

The underlying physics determining the boundary potential is complex, involving repulsive electro-static interactions[33,34] between the plasmid and the cavity boundary and the degree to which the plasmid can be compressed as it is squeezed against the cavity wall. However, we expect that the range of the boundary potential is roughly related to the plasmid size. We determine the interaction range of the boundary potential as the point where it reaches ~$2k_BT$ (corresponding to a suppression of plasmid occupancy of around 90%). Using the fitted values of $r_b$ and $b$, we find that the fitted boundary potential reaches ~$2k_BT$ at a distance of $190 \pm 8$ nm from the cavity boundary. This value is indeed on order of magnitude of the true plasmid size; the radius of gyration $R_g$ of the 5.76 kbp supercoiled plasmid is measured to

**Fig. 6 Dwell time analysis of a cavity confined plasmid in presence of $T_4$-DNA. a** Dwell time histograms for cavities of varying eccentricity with double-exponential fits. **b** Resulting average dwell times extracted from double exponential fits to dwell-time histograms, with the black circles corresponding to the long average dwell-time and the red circles corresponding to the shorter average dwell time. The error bars denote the covariance from the fitting.

be $102 \pm 2$ nm from the light scattering[35]. Note that, in our 10 mM Tris buffer, we expect the DNA effective width $w$ to be around 10 nm[31], while the effective width is closer to 2 nm in the 200 mM NaCl buffer used for the light scattering measurements[35]. As the $R_g \sim w^{1/5}$ for a bulk self-avoiding coil[33], this suggests that the plasmid $R_g$ is closer to 140 nm in our buffer conditions. Regarding the parameter $a$, which determines the magnitude of self-exclusion, we find $a = (1.1 \pm 0.1) \cdot 10^{-6} k_BT \cdot \mu m^3 \cdot bp^{-1}$. The repulsive term of the Flory energy[26] is $v \cdot k_BT \cdot \rho$ where $v$ is the excluded volume and $\rho$ is the polymer solution concentration defined by Kuhn monomer. The persistence length $P$ of the DNA chain is 50 nm, therefore the diameter $a_k$ of the Kuhn monomer is $a_k = 2P = 100$ nm. The contour length of the $T_4$-DNA is around 60 μm, yielding approximately 600 Kuhn segments for the $T_4$-DNA. After normalizing $\rho$ to the number of the Kuhn segments, and approximating the plasmid as a sphere of radius $r_p$, the value of $a$ gives out $r_p = 70 \pm 3$ nm, again the same order of magnitude with the result from light scattering[35]. To check that our conclusions do not depend on the detailed

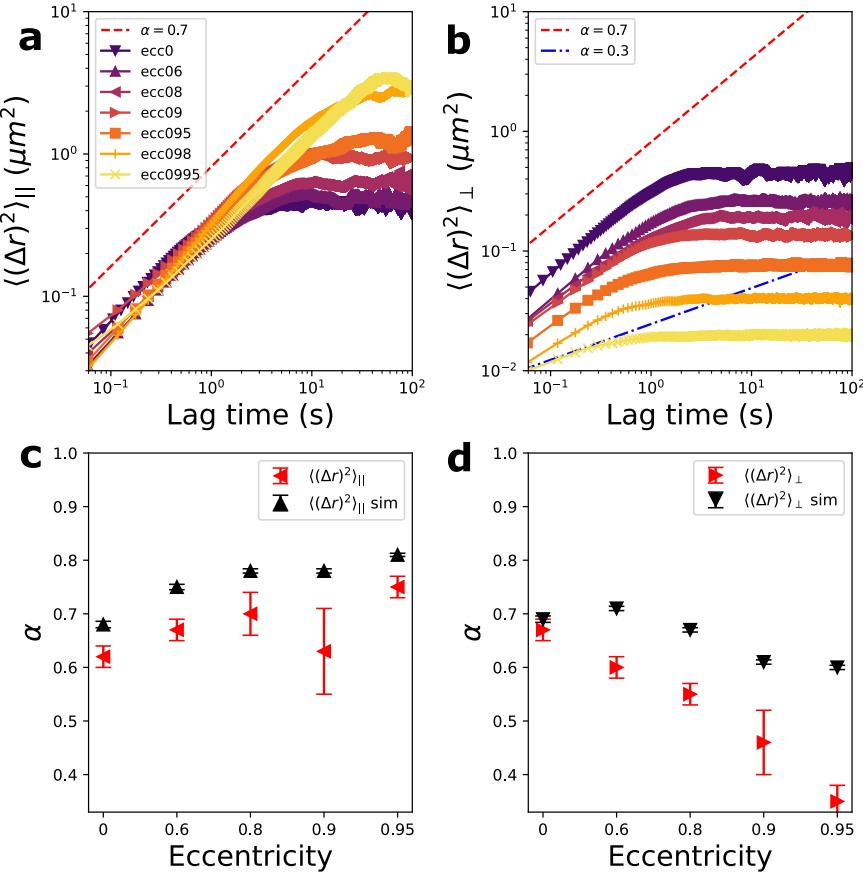

**Fig. 7 Mean-squared displacement of a cavity confined plasmid in presence of T$_4$-DNA. a** The major axis component of the MSD extracted from the plasmid confined in different cavities. **b** The minor axis component of the MSD extracted from the plasmid confined in different cavities. **c** The scaling exponent of $\langle(\Delta r)^2\rangle_\parallel$ extracted from experiments (red points) and simulations (black points). **d** The scaling exponent of $\langle(\Delta r)^2\rangle_\perp$ extracted from experiments (red points) and simulations (black points). The red error bars give the standard error of the mean of $\alpha$ over captured videos ($n = 5$ for $e = 0$, $n = 7$ for $e = 0.6$, $n = 8$ for $e = 0.8$, $n = 9$ for $e = 0.9$, $n = 15$ for $e = 0.95$). The black error bars give the standard error of the mean of $\alpha$ over simulation clips ($n = 250$ clips with 3000 steps).

boundary potential used, we also explored a Weeks-Chandler-Andersen (WCA) form for $u_{\text{wall}}(\mathbf{r}_s - \mathbf{r})$, which rises more steeply than an exponential (see Supplementary Note 4). The WCA model yields similar agreement, reaching $2\,k_BT$ at a distance of $180 \pm 8$ nm, and an excluded volume radius of 60 nm.

**Plasmid dwell time at poles**. The plasmid residence time at the poles characterizes the partitioning stability. The plasmid is considered to be in the pole region when its position satisfies $|x| > l/3$, where $x$ is the major-axis projection of the plasmid position vector and $l$ is the maximum extension the plasmid can reach in the experiment. The histogrammed dwell-time for the various cavities are shown in Fig. 6a. The histogram suggests two different time-scales, which are extracted from a double-exponential model fitting (see Supplementary Note 5). The longer time-scale increases with increasing eccentricity (see Fig. 6b). This time-scale corresponds to the mean dwell time of the plasmid at the cavity poles, and arises from the increased free energy barrier between the cavity pole and cavity waist (located at $\pm l/3$ by our definition of the pole region), also reflected by the reduced preference of the plasmid at the cavity waist in the plasmid position histogram in Figs. 4b and 5a. The shorter time scale arises from events that briefly cross the boundary at $|x| > l/3$ and then return towards the cavity center without experiencing the potential pocket at the poles (see Supplementary Note 5, in particular Supplementary Fig. 6).

**Plasmid mean-squared displacement**. In addition we extract the plasmid's mean-squared displacement (MSD): $\langle(\Delta r)^2\rangle = \langle(\mathbf{r}(t)-\mathbf{r}(0))^2\rangle$ . The MSD projected along the cavity major axis, $\langle(\Delta r)^2\rangle_\parallel$, and minor axis $\langle(\Delta r)^2\rangle_\perp$, are shown respectively in Fig. 7a, b (also see Supplementary Note 6). Note that the saturating value of the MSD reflects the differing spatial extent of the confinement for the different cavities. For the short-time regime, the MSD shows a sub-diffusive behavior ($\alpha < 1$) in both directions. The scaling exponent of the MSD ($\alpha$), determined from a power-law fit to the short-time regime (less than 1 s), is shown in Fig. 7c, d. The scaling exponent for the major axis MSD component increases slightly (Fig. 7c) while the minor axis component strongly decreases (Fig. 7d). In order to understand this behavior, we have performed a Brownian dynamics simulation for a particle undergoing a random walk in a free energy landscape derived from the observed plasmid position distribution (see Supplementary Note 7 for details on simulation methodology). Specifically, we derived the free energy from $F_{\text{CM}}(\mathbf{r}) = -k_BT \log P_{\text{plasmid}}(\mathbf{r})$, where $P_{\text{plasmid}}(\mathbf{r})$ is the fitted probability distribution of the plasmid position in the cavity (i.e., shown in Fig. 5a). We would expect, if the observed MSD behavior results purely from the particular non-uniform structure of the potential landscape, that these simulations would agree with our measurements. We indeed find that the observed trend and exponent values for the MSD major axis component agrees well with simulation (Fig. 7c), suggesting that the sub-diffusive

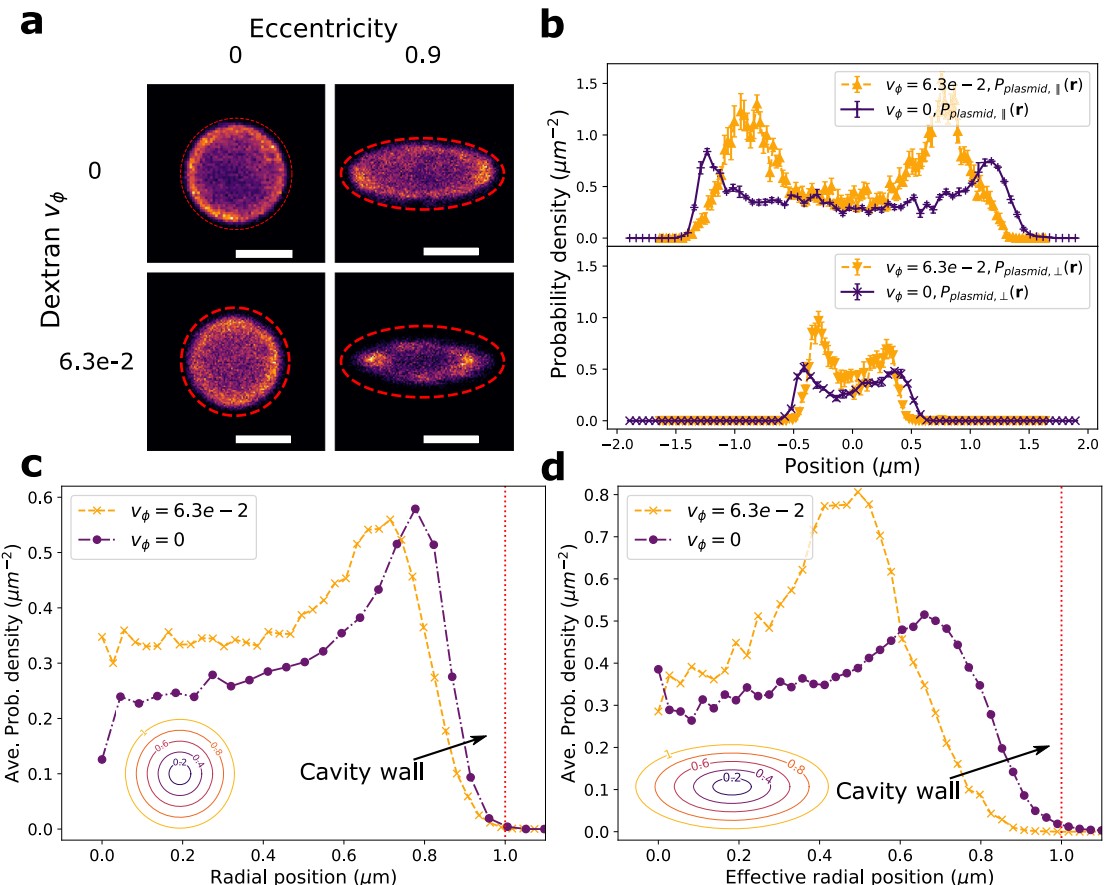

**Fig. 8 Experimental results for a cavity confined plasmid in presence of T₄-DNA with added macro crowders. a** Plasmid probability density in cavity, obtained from histogrammed plasmid position measurements, with and without crowding and for a symmetric and anisotropic cavity. The crowder volume fraction is $v_\phi = 6.3 \cdot 10^{-2}$ and the scale bars are 1 μm. The red dashed line gives the cavity edge. **b** Cross-section of the plasmid position probability along the cavity major axis (upper panel) and along the cavity minor axis (lower panel). The length of the semi-major axis of the cavity is 1.51 μm. The length of the semi-minor axis of the cavity is 0.66 μm. The error bars denote the standard error of the mean of the binned probability density ($n = 3$ bins). **c** Radially averaged plasmid probability density. The inset shows the contour along which the probability density average is taken. **d** Plasmid probability density averaged over an elliptical contour versus effective radial coordinate $r_{\text{eff}} = \sqrt{\frac{x^2}{a^2} + \frac{y^2}{b^2}}$, where $a$ and $b$ correspond to the length of semi-major and semi-minor axis respectively. The inset shows the elliptical contours along which the probability density average is taken.

behavior arises largely from the non-uniformity of the free energy landscape. The MSD minor axis component, however, falls more steeply than the simulation results, with the exponents showing much stronger sub-diffusive behavior (lower $\alpha$, Fig. 7d). This behavior may arise as the plasmid diffusivity is influenced by interaction with the confined T₄-DNA, which relative to the plasmid, acts effectively like a section of a larger polymer solution. Small particles are well-known to exhibit a size-dependent anomalous diffusion in polymer solutions, i.e., departing from pure Stokes-Einstein diffusion, as the polymer solution can give rise to a non-continuum resistance at scales on order of the particle size[36–39]. In particular, the particle diffusivity decreases as a function of polymer volume fraction[37–39], and sub-diffusive regimes have been observed[37,40]. We suggest that these anomalous effects are observed for the minor axis, and not major axis MSD component, as the MSD minor axis component is sensitive to the plasmid's motion through regions of concentrated DNA along the central cavity axis (i.e., as occurs when a plasmid makes a perpendicular crossing across the cavity major axis). Note that we observe an absence of super-diffusive behavior ($\alpha > 1$), suggesting that super diffusion does not arise purely from polymer entropic and elastic recoiling forces, as hypothesized in Hsu et al.[10]

**Effect of macromolecular crowding on plasmid distribution.** We introduce small inert molecules (dextran, gyration radius ~2.6 nm) into the plasmid–T₄-DNA confinement system to simulate the effect of molecular crowding. We observe that a high concentration of dextran (volume fraction $v_\phi = 6.3 \cdot 10^{-2}$) alters the plasmid probability density in a manner that depends on the overall cavity anisotropy (Fig. 8a). Specifically, for circular and anisotropic cavities, crowders displace the plasmid probability density inwards from the cavity edges while also enhancing segregation of plasmids from the cavity center towards the cavity edges. We quantify the observed phenomenon by measuring, for a circular cavity, a radially averaged plasmid probability density (Fig. 8c) and, for the anisotropic elliptical cavity, the plasmid probability density along cross-sections parallel and perpendicular to the cavity major axis (Fig. 8b) as well as the plasmid probability density averaged along an elliptical contour (Fig. 8d). We plot the plasmid probability density averaged over the elliptical contours versus an effective radial coordinate defined as $r_{\text{eff}} = \sqrt{\frac{x^2}{a^2} + \frac{y^2}{b^2}}$, where $a$ and $b$ correspond to the length of semi-major and semi-minor axis respectively. For purposes of quantifying shifts in the plasmid probability density, the distribution edge is defined as the position where the average plasmid

probability density (in Fig. 8c, d) is equal to $\frac{1}{e}$ of its maximum. The effect of crowding is small for the symmetric cavity; here the plasmid probability densities with and without dextran are qualitatively similar, with only a small enhancement at the cavity edges present (Fig. 8c). For the symmetric cavity, crowders displace the plasmid probability density inward by $0.04 \pm 0.01\,\mu m$. However, the presence of anisotropy ($e = 0.9$) strongly enhances the effect of crowding. In the anisotropic cavity, the inwards displacement is a factor of four greater ($0.16 \pm 0.01\,\mu m$ from the probability density averaged over elliptical cross-section, for comparison displacement along the minor axis is $0.12 \pm 0.03\,\mu m$ and the displacement along major axis is $0.18 \pm 0.03\,\mu m$). Additionally, the segregation effect, towards both the cavity periphery and poles, is amplified (Fig. 8a, d).

Neutral dextran nanoparticles are expected to influence the system purely entropically[41–43]. While crowders promote the compaction of $T_4$-DNA, yielding more space accessible to the plasmid, the crowders can also accumulate at the cavity perimeter[42], reducing the accessibility of the cavity edge. The plasmids then tend to occupy an intermediate region between the cavity edge, with its high concentration of crowders, and the central region of the cavity occupied by the $T_4$-DNA. The increased overall compaction of the $T_4$-DNA increases the $T_4$-DNA concentration in the cavity center and thus the influence of excluded volume, enhancing the segregation effect of the plasmids relative to the $T_4$-DNA. Critically, anisotropy enhances the effect of crowding (in Fig. 8a, eccentricity = 0.9). One possible explanation is that the chain conformation itself has a transient anisotropy[44] with increased elongation along a particular axis. In the symmetric cavity, this elongated axis can rotate freely, so that the plasmid and the crowders, which are much smaller than $T_4$-DNA, can maximize their accessible volume by moving in a coordinated fashion with the $T_4$-DNA (transiently occupying regions to either side of the elongated chain). However, in the anisotropic cavity, the more compact $T_4$-DNA aligns with the cavity major axis and is rotationally constrained. The crowders are thus forced to accumulate preferentially towards the cavity edge.

## Discussion

In conclusion, using a nanofluidic model system, we demonstrate that anisotropic confinement can give rise to polar organization of a two-polymer system due to entropy-driven chain demixing. In our first experiment, we observe that as the cavity aspect ratio is increased, the two DNA chains will transition from a rotationally symmetric state that lacks polar ordering to a polarized state with a polar alignment of the chain center-to-center vector. In our second experiment, we observe that when a large DNA molecule is confined in an anisotropic cavity in the presence of a plasmid, the combination of excluded volume interactions and repulsive interactions with the cavity boundary will lead to the plasmid adopting a polar preference. These experiments illustrate physical principles that may play a role in more complex phenomena in bacteria. The first experiment shows how entropy-driven chain demixing can segregate two equal-size molecules in anisotropic confinement, which has been proposed as possible mechanism promoting chromosomal segregation in bacteria. The second experiment illustrates a principle that may help explain the observed distribution of high-copy number (hcn plasmids, >15 copies per cell) plasmids.

To expand on the second point, plasmids present at low-copy number (lcn) possess dedicated molecular machinery for ensuring proper partitioning upon bacterial division. In contrast, active partitioning mechanisms are not known to exist for plasmids present at high copy number[45]. While purely random partitioning can theoretically ensure stable transmission in the case of high copy number, in vivo imaging of fluorecently labeled plasmids in *E. coli* suggests that hcn plasmid partitioning is not random. In particular, the live-cell work suggests that the hcn plasmid distribution has a remarkable multi-focal character, with large multi-plasmid clusters present at the cell poles[8]. Observation of anti-correlation between nucleoid location and plasmid distribution suggest that this polar organization arises from nucleoid occlusion, i.e., the plasmids are physically obstructed from nucleoid proximal regions in the bacteria mid-section[8]. Super-resolution studies support but complicate this picture, indicating that the excluded plasmids are in fact roughly distributed in a ring around the nuceloid periphery, with a small degree of nucleoid penetration[9]. Recent polymer-based simulations confirm that entropic forces will tend to segregate plasmid and chromosomal dsDNA, but predict that the exclusion is strongly size dependent, with larger plasmids (>100 kbp) tending to occupy the cell poles, and smaller plasmids excluded laterally about the nucleoid without showing pronounced polar organization[11]. Our measurements support the conclusion that the peripherial ring distribution and polar clusters arise from generic features of the entropy-driven interactions between large polymer chains, but go further in suggesting that the polar organization results from competition between excluded-volume interactions and repulsion from the anisotropic confining surfaces. Our findings additionally suggest that molecular crowding may influence the degree of plasmid segregation and polar accumulation and that the effects of crowding are enhanced by cavity anisotropy.

Note that, while bacterial chromosomes are ~Mbp in scale (e.g., the genome size of *E. coli* is 4.6 Mbp), much larger than the DNA sized used here ($\lambda$-DNA is 46.5 kbp, $T_4$-DNA is 166 kbp), the chromosomal systems do not necessarily contain more independent chain units, due to their high degree of compaction arising from negative supercoiling and associated proteins. The structural unit of a chromosome is estimated to be between 10 and 300 kbp in size, giving rise to between 15 and 400 structural units[16]. In comparison, a bare $\lambda$-DNA and $T_4$-DNA molecule, for which the structural unit is the Kuhn length (100 nm or 300 bp), has respectively 145 and 500 structural units. Thus, due to the chromosome's strong degree of compaction, the chromosomal polymer model and the simple DNA model have an effective polymer size at least order of magnitude comparable. Note, however, that the more anisotropic chain unit in the simple DNA model may lead to subtly different scaling behavior of the chain free energy as our nanofluidic model may technically lie in an extended confinement regime[16,46].

Our nanofluidic system, with a 200 nm height significantly smaller than the width (~1 $\mu m$), is slit-like. Jun et al. has suggested that slit-like systems should show enhanced segregation relatively to isotropic systems[6], which is consistent with what we observe; even in a circular cavity, the $\lambda$-DNA molecules do not instantaneously mix. However, note that there exist real biological systems that resemble our slit-like cavities. For example, *H. walsbyi*, an archea that is found world-wide in brine pools, has a stamp-like shape with a thickness of less than 0.2 $\mu m$ and a width around 2–5 $\mu m$[47]. During cell growth, *H. walsbyi* transforms from a square into a rectangular shape; this may induce anisotropic confinement that helps ensure chromosome partitioning prior to division[48].

From the point of view of simulation, model experimental systems, which contain a precisely calibrated degree of complexity, can help validate/calibrate simulation approaches[49] and thus serve as a stepping stone to modeling the full complexity of an in vivo biological system. In particular, our experiment gives us access to time-scales associated with the two-chain polymer

dynamics. These time-scales can be challenging to access in simulations as computational approaches that can capture dynamics require unfeasibly long simulations times to model chains of a size approaching that of the chromosomal polymer models[49]. In the λ-DNA experiment, we observe a time-scale ~10 s associated with molecule pole-swapping. The existence of this time-scale, which is much smaller than an overall bacterial generation time (>20 min[50]), suggests an additional role for mechanisms that ensure anchoring of replication origins to cell poles[51] (i.e., entropic mechanisms may not be sufficient to ensure stable polar partitioning of chromosomes due to pole-swapping events). In the plasmid system, measurements of the MSD from single plasmid trajectories suggest that the observed scaling exponents are consistent with sub-diffusion, and that the observations of super-diffusive exponents in live cells[10] likely result from additional active mechanisms rather than entropic forces.

Our nanofluidic model permits a wide-range of additional experiments that can, following a "bottom-up philosophy"[52], explore the global significance of additional biological complexity on the overall entropy driven chain demixing. Our addition of crowding agents to the plasmid–T₄-DNA model is an example of how we can increase system complexity step-by-step. Additionally, we could use circular DNA constructs to explore the role of circular chain topology. We could add variable numbers of plasmids and DNA molecules of varying size to simulate secondary chromosomes and different degrees of plasmid loading. The simple DNA constructs used here could be potentially replaced with extracted bacterial nucleloids[43] and the role of specific nucleoid associated proteins explored (e.g., H-NS and Fis proteins[53]).

A second potential application of our system is to attempt to elucidate certain poorly understood in vivo phenomena that appear impacted by nucleoid exclusion. In particular, the formation of localized aggregates of unfolded/mis-folded protein is a widespread phenomenon in bacteria[54]. These aggregates often appear at the cell-poles (as in the case in *E. coli*[54]), and form in response to proteotoxic stresses arising from cellular and environmental factors, for example decline in ATP levels[54], heat shock[55], antibiotic treatment, high levels of heterologous protein expression[56] and potentially cell aging[57] (although the correlation of aggregate formation with cell aging is under debate[58]). The aggregates are inheritable and associated with increased with increased resistance to stress[55] with a close connection to persister phenotypes that can survive starvation and exposure to high levels of antibiotics[56,59,60]. The aggregates have been observed to freely diffuse in nucleoid-free regions of the bacteria[61,62], suggesting that entropic forces may play a role in their polar localization, analogous to the localization of the plasmids. In principle, protein aggregates, for example extracted from bacteria via centrifugation[63] and labeled via IbpA-YFP fusion proteins[55], could be introduced to our nanofluidic system to observe if polar organization occurs in the absence of any active mechanisms. Protein aggregates in non-stressed conditions often form at only one of the cell poles[57]. We suggest this may result due to competition between protein aggregates and other macromolecular components, such as plasmids, for polar locations (a hypothesis our nanofluidic model allows us to partially test, by exploring systems containing mixtures of plasmids and proten aggregates). Lastly, given that the geometry and molecular constituents of these model experiments are precisely known, the results can then be compared directly against molecular simulation, which will enable calibration of simulation predictions regarding the role of specific biological features.

## Methods

**Nanofluidic device and interface chuck**. The device fabrication protocol is based on the procedure described in Capaldi et al.[21] Briefly, contact photolithography

and RIE is used to define a nanoslit on a borosilicate glass wafer; electron beam lithography followed by RIE defines the geometry of each elliptical cavity. We use an RIE recipe based on etch parameters suggested by Goyal et al.[64] with which we can produce smooth borofloat surfaces with $R_a$ less than 5% of the etching depth. The etched borosillicate wafer is then anodically bonded to a silicon wafer containing a 100 nm LPCVD silicon nitride film (ordered from the Cornell Nanofabrication facility). The silicon is then etched in KOH solution to reveal the nitride membrane at the device center. See Supplementary Note 8 for a process flow chart.

**Two-color fluorescent microscopy and pneumatic pressure control**. The λ-DNA (48.5 kbp, linear topology) is stained with YOYO-1 and YOYO-3, T₄-DNA (169kbp, linear topology) is stained with YOYO-3, pBR322 (4361 bp, ring topology) is stained with YOYO-1. The staining ratio is controlled to 10:1 bp:fluorophore. The analytes are diluted to 2.5 μgmL⁻¹ in 10 mMol Tris (8.0 pH). Betamercaptoethanol (BME) 2% by volume is added to sample solution prior to experiments to reduce photobleaching and photonicking. In order to access the nanofluidic device optically and pneumatically, we mount the device on a chuck fabricated by a stereolithography 3D printer from Formlabs with Formlabs standard clear resin (using 25 μm resolution). The chuck is submerged in an IPA bath for 3 min after printing and the access channels are flushed manually with IPA to prevent blockage. A 30 min UV post-curing at 60 °C is applied to harden the chuck and remove the IPA residue. The chuck is interfaced to the device with a customized rubber sheet gasket and the device is secured onto the chuck by a stainless steel face plate. The chuck is then mounted on a Nikon Eclipse Ti inverted microscope with a Nikon Plan Apo VC 100x oil-immersion objective and an Andor iXon X3 EMCCD camera. The imaging system is controlled by the open-source software μ-Manager. In order to perform two-color fluorescence imaging, we developed an LED based two-color excitation system triggered externally by the exposure signal (this system is developed and released as an open-source project[65]). The nitride membrane is actuated by interfacing a nitrogen gas controller to the membrane window via a luer connector. The pneumatic pressure is modulated by a benchtop pressure controller, which communicates with the computer via a NI DAQ board using a homemade NI LabVIEW program. The DNA sample loading is controlled manually by a syringe pump. The inlet and outlet of fluidic channel are connected directly to the atmosphere when the membrane is actuated. A constant 1500 mbar pressure is applied to ensure the membrane is completely deflected and the videos are captured in real-time. For two λ-DNA chains confinement experiment, 2 videos are captured from 2 cavities with $e = 0$ on 1 chip; 6 videos are captured from 3 cavities with $e = 0.3$ on 1 chip; 10 videos are captured from 2 cavities with $e = 0.6$ on 1 chip; and 9 videos are captured from 4 cavities with $e = 0.9$ on 1 chip. For T₄-DNA and plasmid DNA confinement experiment, 5 videos are captured from 5 cavities with $e = 0$ on 2 chips; 7 videos are captured from 4 cavities with $e = 0.6$ on 1 chip; 8 videos are captured from 5 cavities with $e = 0.8$ on 1 chip; 9 videos are captured from 5 cavities with $e = 0.9$ on 1 chip; 15 videos are captured from 13 cavities with $e = 0.95$ on 1 chip; 9 videos are captured from 9 cavities with $e = 0.98$ on 1 chip; and 16 videos are captured from 10 cavities with $e = 0.995$ on 1 chip. Each video contains a trapping event and the duration of each video is ~5 min depending on the photobleaching.

**Image analysis**. We subtract background noise prior to the analysis using a noise subtraction algorithm proposed by Tang et al.[66] implemented in ImageJ. The fluorescence center of mass (FCM) is then calculated for each frame via:

$$\mathbf{r}_{\mathrm{CM}}(t) = \frac{\sum \mathbf{r}(t) I(\mathbf{r}, t)}{\sum I(\mathbf{r}, t)} \tag{2}$$

The position data is then fed to a homemade open-source Python script to perform the remaining analysis (e.g., calculation of cavity probability distributions, free energy).

**DNA-dextran sample preparation**. Dextran with molecular weights $M_w = 5$ kDa was purchased from Sigma-Aldrich and dissolved in 10 mM Tris buffer. The tris-dextran buffer is mixed with plasmid–T₄-DNA containing buffer in equal volumes and incubated for 48 h with 2%v/v of BME added right before the experiments. The radius of gyration of the dextran molecules is calculated to be 2.6 nm via the empirical equation $R_g = 0.066 \cdot M_w^{0.43}$ with $M_w$ in g/mol and $R_g$ in nm[42].

**Reporting summary**. Further information on research design is available in the Nature Research Reporting Summary linked to this article.

## Data availability
The data that support the findings of this study are available from the corresponding author upon reasonable request. Source data are provided with this paper.

## Code availability
The codes for data analysis is available at https://github.com/echolzz/nanocavity_coderepo.

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

## Acknowledgements

Funding is provided by Natural Sciences and Engineering Research Council of Canada (NSERC, Grant no. RGPIN-2018-06125) and the Fonds de rechercheé du Québec-Nature et technologies (FRQNT, PR-208174). The authors also thank LMN facility at INRS-Varennes, McGill Nanotools-Microfab and Facility for electron microscopy research (FEMR).

## Author contributions

Z.L. and W.R. conceived the idea. Z.L. and X.C. fabricated the chips. Z.L. performed the simulation with L.Z.'s help. Z.L. designed and conducted the experiments with Y.Z.'s help. R.R.L. assisted in the project/experiment design. Z.L. performed the data analysis. Z.L. and W.R. wrote and revised the manuscript. W.R. and Z.L. managed the project. All authors contributed to discussions and paper revisions.

## Competing interests

The authors declare no competing of interests.
