## [Peer Review File · Nature Communications]

Confinement Anisotropy Drives Polar Organization of Two DNA Molecules Interacting in a Nanoscale CavityReviewers' Comments:

Reviewer #1:

Remarks to the Author:

This manuscript describes experiments on fluorescently labelled DNA molecules, both relatively long linear λ - or T4-DNA and a smaller circular plasmid, confined to shallow elliptical microcavities of varying eccentricities. The authors argue that idealized experiments such as these, focusing on the physics of confined polymers, are indispensable in unravelling the intricacies of chromosome segregation in bacteria.

The major findings are:

- that with increasing eccentricity the spatial separation of two linear chains along the major increases.
- two linear chains can swap position with an eccentricity-dependent dwell time of the order of 10-100 s.
- plasmids are expelled from the linear chains and with increasing eccentricity become more localized to one of the poles of the confining ellipse.

Arguably, these are first experimental results of this kind, and as such merit attention. However, at the same time, the results are also somewhat disappointing:

- From a purely physical perspective, the outcomes are hardly surprising. Over the past decade and a half, the mechanisms of polymer organization in strong confinement have extensively been studied, albeit through simulations. The reported observations of the shape-dependent organization of either two polymers or a polymer and a plasmid are inevitable consequences of the enhanced entropic repulsion under strong confinement.
- From a biological perspective, this study fails to confront the most pressing issue in this field, viz. do two initially intermingled polymers spontaneously segregate on a reasonable time-scale under strong confinement (see e.g., Minina & Arnold, *Soft Matter* (2014), 10: 5836-5841). By design the polymers in the experiments presented are demixed in bulk solution to begin with and subsequently confined to the chambers in the already demixed state. As such they describe only the properties of the final state, but not the all-important segregation process itself.

Minor point:

- To the mind of the reviewer the analogies drawn in several places with liquid crystalline ordering are spurious. Liquid crystallinity is an emergent effect of the interaction between anisometric molecular moieties that allow the definition of a persistent molecular frame to describe their individual orientation. The manuscript deals with the anisotropic spatial distribution of otherwise effectively isotropic molecules which is imposed by the anisometric boundary conditions.

Reviewer #2:

Remarks to the Author:

The manuscript of Liu et al. presents an experimental study of the behaviour of two DNA molecules confined inside cavities of different aspect ratio. The study builds on a setup for nanoscale confinement of differentially staining DNA molecules that has been previously validated by some of the authors (Capaldi et al., *Soft Matter* 2018).

In this study, the general setup is used to track how two DNA molecules of equal or different size behave when the confinement is made increasingly one-dimensional- or channel-like.

The study is carried out with rigour, is scholarly presented, and conveys an unprecedented wealth of quantitative detail on the entropic segregation of molecules in low-dimensional confinement. This is a well-established research topic, for which there have been numerous theoretical, numerical contributions. There have been several experimental investigations too, but not with the type of control on confinement degree and effective dimensionality of the present one, nor the amount of

detail on the positioning of the two chains both relative to each other and relative to the confining well.

I am thus happy to recommend the article for publication in Nature Communications.

In my view, the main point that deserved a better discussion, and possibly an improved data analysis, regards section E, that is the scaling analysis of the plasmid mean-square displacement along major and minor axes of the ellipsoidal cavity.

It looks to me that the projected motion of the plasmid ought to be significantly affected by the width in the transverse direction. My suggestion would be to take into account with a simple model of diffusion in a potential well with profile given by $-\log(w)$, where w is the transverse width. This ought to better represent the motion of the plasmid in the uniform background of the cavity and would allow to better establish the intrinsic diffusivity exponent α that provides the best match to the projected MSD,

Minor points:

- Fig. 7, first two lines of the caption. Should "minor" and "major" be swapped?

- in lines 520-530, the authors elaborate on the expected transferability of the present conclusions to much larger systems, such as chromosomes. While I believe in the generality of the present results, I am not sure about the validity of the mapping that the authors put forward in terms of equivalent number of "chain units". The nature of the chain units, or effective monomers, is very different across the quoted examples, and the equivalent polymer chains would have very different width to length ratio and renormalised persistence lengths too. These are all factors known to affect the metric scaling regimes of the chains. It looks to me that the argument in terms of chain units, which I did not find necessary in the context of the manuscript, may not hold.

Reviewer #3:

Remarks to the Author:

Liu et al. use either two DNA molecules or one DNA molecule and a plasmid to investigate polymer segregation under confinement. Although interesting, in my opinion the current manuscript has a major issue: their model system can not be used to recapitulate biological systems.

Major issue:

The authors justify their study by stating that their "drastically simplified model system..." allows "...to reproduce features of the in vivo systems." I believe that this is too bold a statement and that the author's data do not justify this statement.

As the authors acknowledge, the prokaryotic intracellular space is densely packed with thousands of different molecules, not only DNA, but also RNA, proteins, metabolites etc. This complicated intracellular space can not be recapitulated via two DNA molecules in a microslit. For example, recent evidence ([doi:10.1016/j.molcel.2018.10.022](https://doi.org/10.1016/j.molcel.2018.10.022); <http://dx.doi.org/10.1098/rstb.2018.0442>; <https://doi.org/10.1101/2021.02.15.431274>) suggest that misfolded proteins accumulate at the bacterial poles. How will this impact the DNA demixing reported by these authors? I believe that the authors need to increase the complexity of their system before drawing conclusions about polymer demixing within living organisms. For example, they could carry out similar studies but in bacteria where it is feasible to introduce plasmids labelled with different fluorophores. Alternatively, they could introduce several labelled molecules (DNA, RNA, proteins etc) in their synthetic system.

Beyond the major issue above, the authors findings might be very interesting for a more "physics audience", however, the manuscript would be then better suited for a more "physics journal".

Minor points:

- The text, especially in the introduction, is some times fragmented; it would help if the authors could better link some of their sentences
- They should clarify in the introduction how do they define confinement anisotropy because by looking at Fig. 1a it seems to me that going from left to right confinement is increasing rather than the anisotropy in confinement.
- "When cavity eccentricity is introduced, the rotational symmetry is broken, and the molecules spend more time at the ellipse poles." Is it really the breaking of rotational symmetry that makes DNA spend more time at the poles? Or are the two molecules simply trying to avoid each other while under increasing confinement?
- "The cavity devices are etched 200 nm deep and have a maximum diameter that ranges from 2 μ m (for $e = 0$) to 3 μ m (for $e = 0.9$)." The authors should talk about microslits then

Response to Reviewers

I. INTRODUCTION

We thank all reviewers for their detailed and informative comments. Overall, while we are quite willing to extend this project with additional experiments, we face a challenging situation in that reviewers 1 and 3 are asking for rather different experiments: reviewer 1 proposes to observe spontaneous segregation of initially intermingled polymers; reviewer 3 proposes to add more biologically relevant complexity. These two requested experiments are both interesting and worthwhile. Yet, we can't do both within the context of the current manuscript. Even if we were successful in both cases, combining the new results into one manuscript along with the older results would lead to a product that is too long and unfocused. We appreciate that this may mean the work isn't suitable for *Nature Communications*. However, it does seem that some aspects of this work were attractive to the reviewers. Thus, we have decided, in this reply, to see if we can create a sort of compromise among the reviewers and editors at what is a reasonable plan of attack for revisions to suitably increase the impact in a way that best preserves the overall themes of our initial manuscript.

Note that the "plan of attack" we propose in the replies is based more along the lines of reviewer 3 rather than reviewer 1 (we propose to add additional complexity to the plasmid positioning experiments). This isn't a comment on the interest of reviewer 1's suggestion (we think it is a very interesting suggestion), but more on which proposal is overall most consistent with the manuscript's original focus.

II. REVIEWER 1

This manuscript describes experiments on fluorescently labelled DNA molecules, both relatively long linear λ - or T_4 -DNA and a smaller circular plasmid, confined to shallow elliptical microcavities of varying eccentricities. The authors argue that idealized experiments such as these, focusing on the physics of confined polymers, are indispensable in unravelling the intricacies of chromosome segregation in bacteria.

The major findings are: (1) that with increasing eccentricity the spatial separation of two linear chains along the major increases, (2) two linear chains can swap position with an eccentricity-dependent dwell time of the order of 10-100 s and (3) plasmids are expelled from the linear chains and with increasing eccentricity become more localized to one of the poles of the confining ellipse.

Arguably, these are first experimental results of this kind, and as such merit attention.

Thank you for the positive comments.

However, at the same time, the results are also somewhat disappointing: - From a purely physical perspective, the outcomes are hardly surprising. Over the past decade and a half, the mechanisms of polymer organization in strong confinement have extensively been studied, albeit through simulations. The reported observations of the shape-dependent organization of either two polymers or a polymer and a plasmid are inevitable consequences of the enhanced entropic repulsion under strong confinement.

One consideration is that we feel it is unwise for the field to rely purely on simulation to probe phenomena related to polymer organization. For example, there is always a need for model experimental systems (e.g. like our nanofluidic compartments) to validate theoretical expectations.¹ These model experimental systems, which contain a precisely calibrated degree of complexity, can serve as a stepping stone to modeling the full complexity of an *in vivo* biological system. More subtlety, even when a simulation methodology is validated, model experiments are also critical for proper calibration of simulation models, so as to ensure the simulations can truly provide quantitative results. For example, simulation and scaling approaches may give correct qualitative conclusions regarding the probable effect of entropic segregation on equilibrium chain conformation. However, to ensure that quantitative aspects are also correctly predicted—such as the detailed shape of the free energy landscape and the exact positions of segregation phase boundaries—experiments are needed for proper benchmarking of simulation.

In addition, it is important to be realistic with regards to the capabilities of modern simulation methodologies. There have been significant advances in the ability to simulate the *equilibrium* properties of large polymer systems via Monte Carlo approaches (albeit these simulations are time-consuming and the results cannot be trusted

at a quantitative level without careful calibration against experiments²). However, simulations are still unable to easily capture dynamic properties of large chains, due to the vast CPU time required,¹ and this is without including physically realistic effects like hydrodynamic interactions. Practically, this means that it is difficult for simulations to predict the timescale of dynamical phenomena,³ for example the pole swapping and the dwell times of the plasmid at the poles. The timescale obtained from our experimental system will help calibrate future dynamical simulations and will assist in generalizing the finding from simulations to biological systems.

We have made some expanded comments on this issue in the discussion: “From the point of view of simulation, model experimental systems, which contain a precisely calibrated degree of complexity, can help validate/calibrate simulation approaches¹ and thus serve as a stepping stone to modeling the full complexity of an *in vivo* biological system. In particular, our experiment gives us access to time-scales associated with the two-chain polymer dynamics. These time-scales can be challenging to access in simulations as computational approaches that can capture dynamics require unfeasibly long simulation times to model chains with a size approaching that of the chromosomal polymer models.¹”

Finally, we do agree with the reviewer that entropic repulsion plays a critical role in the shape-dependent organization for both systems. However, we point out that other interactions we identify are also crucial: for example the confinement geometry may play a non-trivial role in the biological process via an interplay with entropic repulsion. To be more specific, in our plasmid-T₄ experiment, the competition between repulsive entropic interactions and interactions with the confining potential creates a pocket structure near the poles, leading to the polar segregation of the plasmids. This is a significant finding that we believe is entirely novel, even in the context of past simulation work. This finding is surprising to us: it is not in our view obvious that the interplay of anisotropic confinement and entropic repulsion alone is sufficient to produce these pockets and hence generate polar segregation.

From a biological perspective, this study fails to confront the most pressing issue in this field, viz. do two initially intermingled polymers spontaneously segregate on a reasonable time-scale under strong confinement (see e.g., Minina & Arnold, Soft Matter (2014), 10: 5836-5841). By design the polymers in the experiments presented are demixed in bulk solution to begin with and subsequently confined to the chambers in the already demixed state. As such they describe only the properties of the final state, but not the all-important segregation process itself.

We agree that the reviewer identifies a very interesting problem. However, we feel that an experimental study of the segregation process of two intermingled chains, while it would be very significant, is beyond the scope of our present work, which focuses on the role of the confinement anisotropy in chain organization. To be clear: this is not to disrespect the reviewer’s suggestion, which is excellent—this is an experiment we have discussed and would like to do—but simply to state that we feel it is truly a topic that would fit more naturally in another paper.

The question then becomes whether the reviewer would view other aspects of the study as sufficiently impactful in principle (although with the understanding that the impact might need to be extended with additional experiments, as per reviewer 3). The problem of plasmid organization is also very pressing, with possible implications for how bacteria may retain or lose immunity to antibiotics.⁴ In particular, the plasmid organization problem has attracted considerable attention with a number of recent *in vivo* studies that track fluorescently labeled plasmids in live cells.⁴⁻⁶ Thus, we submit that our finding that we can generate an annular and polar organization of the plasmid position in our nanofluidic model, resembling that of the biological system, is significant. The reviewer might view the two equal sized chain experiments presented in the article as a sort of ‘warm-up,’ which confirms yet adds quantifying detail to existing expectations from simulations, and then view the plasmid work as an extra (surprising?) finding that carries the bulk of the work’s impact. Perhaps in this context there would be a need to extend the plasmid work with additional experiments (e.g. multiple plasmids, vary plasmid size), which would also target the concerns raised by the third reviewer.

Minor point: To the mind of the reviewer the analogies drawn in several places with liquid crystalline ordering are spurious. Liquid crystallinity is an emergent effect of the interaction between anisometric molecular moieties that allow the definition of a persistent molecular frame to describe their individual orientation. The manuscript deals with the anisotropic spatial distribution of otherwise effectively isotropic molecules which is imposed by the anisometric boundary conditions.

We agree that our system is not closely related to the liquid crystalline ordering phase (i.e. a physical analogy is inappropriate). However, we think that the quantity $S = 2(\cos^2(\theta) - \frac{1}{2})$, which describes the molecular alignment of the liquid-crystal phase, can also serve as an order parameter that quantifies the degree of alignment in our two-chain system. That is to say, while a physical analogy between the two systems is inappropriate, it does

make sense to define the order parameter for our orientational transition in a similar way as for liquid crystals.

Thus, in the revised manuscript we remove text that indicates there is a physical analogy between our system and liquid crystals, but retain the previous definition of the order parameter. In the introduction we omit the line “qualitatively similar to a liquid crystal isotropic to nematic transition” and write instead: “This polar organization of the two chains at the ellipse poles can be understood as an orientational configuration transition that can be quantified by an order parameter analogous to that used for a liquid crystal isotropic to nematic transition.” In section IIA, we remove the clause, “analogous to the director vector in liquid crystals,” the sentence now reads: “The separation vector $\mathbf{r} = \mathbf{r}_1 - \mathbf{r}_2$ serves as an additional measure of symmetry breaking, tracking the self-alignment of the two-chain system along the cavity long-axis.” In the first paragraph of the conclusion, we remove the text “nematic like” and replaced it with “polar.”

III. REVIEWER 2

The manuscript of Liu et al. presents an experimental study of the behavior of two DNA molecules confined inside cavities of different aspect ratio. The study builds on a setup for nanoscale confinement of differentially staining DNA molecules that has been previously validated by some of the authors (Capaldi et al., Soft Matter 2018). In this study, the general setup is used to track how two DNA molecules of equal or different size behave when the confinement is made increasingly one-dimensional- or channel-like. The study is carried out with rigour, is scholarly presented, and conveys an unprecedented wealth of quantitative detail on the entropic segregation of molecules in low-dimensional confinement. This is a well-established research topic, for which there have been numerous theoretical, numerical contributions. There have been several experimental investigations too, but not with the type of control on confinement degree and effective dimensionality of the present one, nor the amount of detail on the positioning of the two chains both relative to each other and relative to the confining well.

I am thus happy to recommend the article for publication in Nature Communications.

Thank you for your support!

In my view, the main point that deserved a better discussion, and possibly an improved data analysis, regards section E, that is the scaling analysis of the plasmid mean-square displacement along major and minor axes of the ellipsoidal cavity. It looks to me that the projected motion of the plasmid ought to be significantly affected by the width in the transverse direction. My suggestion would be to take into account with a simple model of diffusion in a potential well with profile given by $-\log(w)$, where w is the transverse width. This ought to better represent the motion of the plasmid in the uniform background of the cavity and would allow to better establish the intrinsic diffusivity exponent α that provides the best match to the projected MSD.

The reviewer suggests that the projected motion of the plasmid should be affected by the cavity transverse width. We believe the reviewer is referring to Fig. 7a,c, where we show the MSD of the plasmid along the major axis as well as the extracted MSD scaling exponent. These results suggest that the diffusion of the plasmid along the major cavity axis is not strongly affected by the cavity eccentricity (the reviewer is wondering why). An intuitive explanation is that the width of our cavity, which ranges from around 500 nm to 1 μm , is always considerably larger than the plasmid diameter (~ 100 nm).⁷ The plasmid therefore is NOT under a strong lateral confinement in our cavities. In addition, the plasmid, experiencing entropic repulsion from the T₄-DNA and repulsion from the cavity boundary, tends to move within the potential valley formed between the T₄-DNA, highly concentrated in the cavity, and the cavity edge. When the plasmid travels from one pole to the other, it prefers this ring shape “short-cut” compared with the cavity center, where the T₄-DNA concentration is high and the plasmid excluded. Therefore, the projection of the plasmid MSD along the major axis does not change significantly since the actual degree of confinement the plasmid experiences does not vary greatly with the change of the eccentricity (i.e. in fact the confinement is determined by the potential pocket created by the competition between the entropic repulsion and boundary potential, not necessarily the actual cavity width, Supp. Mat Fig. 3d). We think that the suggested model potential with the profile given by $-\log(w)$ does not suit our experiment well based on the argument above (i.e. the plasmid motion is determined by the combination of the potential arising from the T₄-DNA and the boundary potential).

Minor points: Fig. 7, first two lines of the caption. Should “minor” and “major” be swapped?

Yes. We have changed the caption following the comments. Thank you!

in lines 520-530, the authors elaborate on the expected transferability of the present conclusions to much larger systems, such as chromosomes. While I believe in the generality of the present results, I am not sure about the validity of the mapping that the authors put forward in terms of equivalent number of “chain units”. The nature of the chain units, or effective monomers, is very different across the quoted examples, and the equivalent polymer chains would have very different width to length ratio and renormalized persistence lengths too. These are all factors known to affect the metric scaling regimes of the chains. It looks to me that the argument in terms of chain units, which I did not find necessary in the context of the manuscript, may not hold.

We think that our argument in terms of effective chain size is valid, but we agree with the reviewer that we absolutely cannot insist on any sort of an exact mapping between our simple polymer model and the chromosomal system. One issue, as the reviewer points out, is that the geometry of the monomeric chain unit making up the simple DNA model and the chromosome are quite different. The chain unit in the simple DNA model is the Kuhn length (equal to twice the persistence length); this is anisotropic due to DNA semiflexibility.⁸ In contrast, coarse-grained polymer models of the chromosome system assume an isotropic chain unit, determined by the size of the topological independent domains of the chromosome.⁸ At the scaling level, chromosomal polymer models are treated as flexible self-avoiding chains made up of these isotropic chain units⁸ (although there is a very wide range of uncertainty in the size of the chain units appropriate to a chromosome, as we indicate in the manuscript⁸). We think that our estimates of the total number of chain units in the simple DNA model and polymer models of the chromosome are accurate. The former is just a question of the number of Kuhn segments present for the chains of the size we use and the latter a quotation of estimates already existing in the literature regarding the number of chain units in the chromosomal polymer models.⁸ Thus, in a narrow sense, our statement that “the effective polymer sizes [of the chromosomal polymer model vis-a-vis the DNA model] are at least order of magnitude comparable” is accurate. Our point is simply that due to its high degree of compaction the chromosome should not be viewed as behaving as a self-avoiding chain ten times longer than our simple DNA models.

Yet, it is certainly a valid question to what degree the simple DNA model’s more anisotropic chain unit affects our conclusions. It is true that an anisotropic monomer unit increases the critical size required for a coil to show Flory scaling⁸ (i.e. the scaling behavior expected of a flexible self-avoiding chain with an isotropic chain unit, the behavior expected of a chromosomal polymer model to which we are comparing). Note, however, that in bulk in our buffer conditions a polymer the size of λ -DNA is sufficiently large to be over the cross-over where Flory scaling is expected.⁹ In confinement, the situation is more subtle; self-avoidance has a stronger effect on the coil conformation, but distinct self-avoiding regimes exist for polymers with anisotropic chain units where the chain behavior is described by mean-field models, with self-avoidance playing a still strong but slightly weaker role.^{8,10} In these mean-field regimes, sometimes referred to as ‘extended de Gennes regimes,’ or ‘marginal solution regimes,’ the scalings for the chain gyration radius are the same as classic self-avoiding regimes, but scalings for the confinement free energy are different, with the extended confinement free energy closer to that of an ideal semiflexible chain.^{8,11} It is a rather subtle question whether our model is in a classic or extended confinement regime. The cross-over between the classic and extended regimes occurs at a slit height $\sim P^2/w$ (P persistence length, w effective width).¹¹ With $P = 50$ nm and $w = 10$ nm, this would suggest the critical slit height at which cross-over occurs is ~ 250 nm, so we may technically be in the extended regime (cavities are etched 200 nm deep), although the cross-over is not abrupt, with the free energy scaling continuously varying from ideal behavior to real chain behavior as the channel width approaches $\sim P^2/w$. A relevant question, which we do not believe has been currently addressed, is how the scaling regimes should be altered for multiple polymer systems.

Following this discussion, we have decided to modify the discussion as follows (italics indicates added text):

“The structural unit of a chromosome is estimated to be between 10-300 kbp in size, giving rise to between 15-400 structural units.⁸ In comparison, a bare λ -DNA and T₄-DNA molecule, for which the structural unit is the Kuhn length (100 nm or 300 bp), has respectively 145 and 500 structural units. *Thus, due to the chromosome’s strong degree of compaction, the chromosomal polymer model and the simple DNA model have an effective polymer size at least order of magnitude comparable. Note, however, that the more anisotropic chain unit in the simple DNA model may lead to subtly different scaling behavior of the chain free energy as our nanofluidic model may technically lie in an extended confinement regime.*^{8,11”}

IV. REVIEWER 3

Liu et al. use either two DNA molecules or one DNA molecule and a plasmid to investigate polymer segregation under confinement. Although interesting, in my opinion the current manuscript has a major issue: their model system can not be used to recapitulate biological systems.

Major issue: The authors justify their study by stating that their "drastically simplified model system..." allows "...to reproduce features of the in vivo systems." I believe that this is too bold a statement and that the author's data do not justify this statement.

We agree with the reviewer that this claim is too bold; *this is not in fact the claim that we intended to make.* It appears we inadvertently gave the reviewer the impression that we were making such a drastic claim by a statement in the introduction that was written too broadly.

We absolutely agree with the reviewer that one has to be very careful making claims regarding the degree to which one can "reproduce" the biological system given the highly simplified/artificial nature of the system we are working with. In fact, any kind of statement saying that the model results are "like" the biological system can be read as problematic (reading the word "like" as a kind of logical "equals"). After all, as the reviewer points out, the model system and the biological system are not the same thing at all, due to the vast complexity of the biological system relative to our model, the geometrical differences, etc.

The problematic statement that we believe misled the reviewer, located at the end of introduction, is: "These results highlight the minimal physical ingredients required to reproduce features of the in vivo systems." We feel that this statement is too broad and can easily be misread. Upon reflection, we think that the best solution is to simply remove this statement from the introduction so it does not read as a major claim, and then only in the discussion focus on the relation of this work to biological questions. This will help emphasize the necessarily somewhat tentative nature of these analogies. The language currently used in the discussion seems appropriate to us and consistent with our true intent: for example "These experiments illustrate physical principles that may play a role in more complex phenomena in bacteria" and "The second experiment illustrates a principle that may help explain the observed distribution of high-copy number (hcn) plasmids."

Our objective in these experiments is best expressed by the first sentence of the introduction: "to develop a minimal model to explain how polymer-polymer interaction in anisotropic confinement can give rise to states with polar organization." Our aim is not to recapitulate a biological system, but rather the opposite, to drastically strip away complexity and thereby identify fundamental physical principles that may be behind more complex phenomena in biology.

As the authors acknowledge, the prokaryotic intracellular space is densely packed with thousands of different molecules, not only DNA, but also RNA, proteins, metabolites etc. This complicated intracellular space can not be recapitulated via two DNA molecules in a microslit. For example, recent evidence (doi:10.1016/j.molcel.2018.10.022; http://dx.doi.org/10.1098/rstb.2018.0442; https://doi.org/10.1101/2021.02.15.431274) suggest that misfolded proteins accumulate at the bacterial poles. How will this impact the DNA demixing reported by these authors?

We agree with the reviewer that the complicated intracellular space cannot be recapitulated via two DNA molecules in a microslit. It is certainly the case that the phenomena we investigate may be influenced by the presence of additional biological complexity that is not included in our simple model. For example, the reviewer suggests that accumulation of misfolded proteins at the cell poles might play a role in the DNA demixing phenomena.

After studying the very interesting references provided by the reviewer,^{12,13} we do think that there are compelling biological reasons why the plasmid polar organization relevant to our study is not *caused* by accumulation of misfolded proteins. The main issue is that the misfolded protein accumulation phenomena referred to by the reviewer appears to occur in special circumstances; namely, it is associated with a special phenotype known as the persister state that arises when bacteria are exposed to high levels of antibiotics¹² or are starved.¹³ When cells enter the persister state they cannot grow but achieve a level of antibiotic tolerance; when the antibiotic is withdrawn, the persister cells resume growth. The persister state is associated with accumulation of misfolded proteins at the cell poles, with the degree of cell dormancy related to the degree of protein accumulation. We contend that, as the plasmid segregation phenomena arises under normal growth conditions,⁵ i.e. does not require that the cells be exposed to antibiotics or starved for nutrients, plasmid segregation is not causally linked to polar accumulation of protein.

However, under conditions where the persister state emerges, we do think that there could be an interesting interaction between the protein accumulation phenomena and plasmid segregation. For example, from a physical point of view, we can hypothesize that the localization of misfolded protein arises from physical phenomena (e.g. entropic repulsion) analogous to the localization of the plasmids. The aggregates may reach a critical size and then experience the entropic effects. The protein aggregates and plasmids, both excluded from the nucleoid, then interact. For example, one curious phenomena we note in the references is that the protein aggregates often form at only *one* of the cell poles;^{12,13} could the plasmids be being entropically displaced to the alternate pole?

Indeed, studying the interaction of these conjoined phenomena could be an interesting application of our nanofluidic model. One could add protein aggregates to our system and observe if there is polar organization and plasmid displacement to the alternate pole. Such an experiment would enable one to test the hypothesis that the biological phenomena in question might arise from physical interactions mediated in a confined environment; this is the advantage of our approach. Note that, even in this case, the model system is still drastically simplified compared to the biological system. Even if one does succeed in reproducing the phenomena, this does not *prove* that exactly the same mechanisms are at play in the biological system; there could always be extra components involved *in vivo*. This is of course an intrinsic drawback to drawing analogies from models that necessarily limit system complexity. Our conclusion, however, is not that this makes such model experiments uninteresting, but rather that *they cannot stand alone*, and must complement in *in vivo* work. This issue is shared by any sort of *in vitro* experiment that attempts to isolate part of the biological system complexity and not work on the whole cell (for example, observe the functioning of isolated components of a biochemical network). What makes our *in vitro* system special is that we use nanofluidics to add confinement effects to *in vitro* experiments that would classically be performed in bulk in a test-tube, allowing us to explore polymer interaction phenomena that cannot be investigated in bulk.

I believe that the authors need to increase the complexity of their system before drawing conclusions about polymer demixing within living organisms. For example, (1) they could carry out similar studies but in bacteria where it is feasible to introduce plasmids labelled with different fluorophores. (2) Alternatively, they could introduce several labelled molecules (DNA, RNA, proteins etc) in their synthetic system.

We agree with the reviewer that it is highly worthwhile to increase our system complexity. Regarding suggested experiment (1), we should point out that there is already a long-history of experiments quantifying plasmid organization in bacteria, using exactly the fluorescent labeling methods referred to by the reviewer. In particular, the polar and annular segregation of plasmids about the nucleoid are very well-established in an *in vivo* context (work that we summarize in the discussion).^{5,6,14} We do not think that performing more such experiments would increase the paper's impact. The added value of this study is that we show the demixing phenomena can arise in the context of a drastically simplified model system, identifying minimal physical ingredients required and highlighting the key fundamental physical mechanisms potentially responsible.

Regarding suggested experiment (2), this type of program is very much along the lines of what we would like to do. Following a bottom-up philosophy, we imagine adding complexity in a step-by-step fashion, so that the effect of each additional component can be isolated. Specific experiments that we have considered include: (a) Introducing small inert molecules, like dextran, to simulate the effect of molecular crowding and (b) exploring demixing in systems that involve multiple plasmids with fixed/variable size. However, we were wondering if the reviewer could be more specific regarding the experiments that will enable us to draw valid conclusions regarding polymer demixing. The issue is that our model can of course not reach the full biochemical complexity of a cell. We are not trying to recapitulate this complexity, but rather reduce it, to find the *essential* degree of complexity required to reproduce the observed phenomena. Our experiments indicate that entropic exclusion alone, operating in an asymmetric confined environment between a DNA molecule and plasmid, can indeed produce an annular and polar positioning of the plasmid. Yet, a biologist could argue that there is essential complexity we are missing regarding this phenomena, without which no analogies can be drawn. The question is what is this missing essential complexity? Would the reviewer view experiments (a) and (b) as a step in the right direction?

Beyond the major issue above, the authors findings might be very interesting for a more "physics audience", however, the manuscript would be then better suited for a more "physics journal".

Yes, this article unabashedly takes a physics perspective, but we think that a physics perspective has value in biology! Recent work indicates that biological systems exploit physical as well as active mechanisms (liquid-liquid phase transitions being a classic example). However, direct tests of physical mechanisms in biological systems are challenging *in vivo*. The physical mechanisms often have a global character, involving the coordinated interaction

of many molecules or large macromolecular complexes across the cell, or involve bulk parameters like cell geometry that are difficult to systematically control *in vivo*. Another issue is that physical and active processes may act in tandem or competition, and can be difficult to separate *in vivo*.

A physics approach is to develop a “minimal model” that strips away complexity and aims to identify the essential features of the system driving the observed phenomena. In particular, a minimal model is simple enough that it can be explored over a large parameter space, much larger than is possible from the study of *in vivo* systems that might occupy only a few distinct points in parameter space. While a minimal model may well fail to account for the observed phenomena, it is still successful if it establishes an effective physical baseline (e.g. behavior in absence of ATP-driven active mechanisms). This baseline can then clarify the role of any complementary active mechanisms that entail metabolic cost. This point of view is not so different from the idea of a biologist running an *in vitro* model to examine isolated features of a biological system (e.g. to check if certain components of a biochemical network identified in *in vivo* actually behave as expected, and to shed light on the necessary concentrations of the chemical components required to generate the *in vivo* behavior). Critically, such *in vitro* experiments have a purely *complementary* role; they are not intended to replace or recapitulate in any sense *in vivo* experiments with live cells, but provide additional insight into the *in vivo* observations.

-
- ¹ J. M Polson and D. A Rehel. Equilibrium organization, conformation, and dynamics of two polymers under box-like confinement. *Soft-Matter*, 17:5792–5805, 2021.
 - ² Thomas E Gartner III and Arthi Jayaraman. Modeling and simulations of polymers: a roadmap. *Macromolecules*, 52(3):755–786, 2019.
 - ³ Marissa G Saunders and Gregory A Voth. Coarse-graining methods for computational biology. *Annual review of biophysics*, 42:73–93, 2013.
 - ⁴ Tai-Ming Hsu and Yi-Ren Chang. High-copy-number plasmid segregation—single-molecule dynamics in single cells. *Biophysical journal*, 116(5):772–780, 2019.
 - ⁵ Rodrigo Reyes-Lamothe, Tung Tran, Diane Meas, Laura Lee, Alice M Li, David J Sherratt, and Marcelo E Tolmasky. High-copy bacterial plasmids diffuse in the nucleoid-free space, replicate stochastically and are randomly partitioned at cell division. *Nucleic acids research*, 42(2):1042–1051, 2013.
 - ⁶ Yong Wang, Paul Penkul, and Joshua N Milstein. Quantitative localization microscopy reveals a novel organization of a high-copy number plasmid. *Biophysical journal*, 111(3):467–479, 2016.
 - ⁷ David R Latulippe and Andrew L Zydney. Radius of gyration of plasmid dna isoforms from static light scattering. *Biotechnology and bioengineering*, 107(1):134–142, 2010.
 - ⁸ Bae-Yeun Ha and Youngkyun Jung. Polymers under confinement: single polymers, how they interact, and as model chromosomes. *Soft Matter*, 11(12):2333–2352, 2015.
 - ⁹ Yanwei Wang, Douglas R Tree, and Kevin D Dorfman. Simulation of dna extension in nanochannels. *Macromolecules*, 44(16):6594–6604, 2011.
 - ¹⁰ W Reisner, Jonas N Pedersen, and Robert H Austin. Dna confinement in nanochannels: physics and biological applications. *Reports on Progress in Physics*, 75:106601, 2012.
 - ¹¹ G. K. Cheong, Xiaolan Li, and K. D. Dorfman. Evidence for the extended de gennes regime of a semiflexible polymer in slit confinement. *Phys. Rev. E.*, 97:022502, 2018.
 - ¹² Y. Pu, Jin X. Li, Y. and, T. Tian, Q. Ma, Z. Zhao, S.-Y Lin, Z. Chen, B. Li, G. Yao, M. C. Leake, C-J Lo, and F. Bai. Atp-dependent dynamic protein aggregation regulates bacterial dormancy depth critical for antibiotic tolerance. *Molecular Cell*, 73:143–156, 2019.
 - ¹³ L. Dewachter, C. Bollen, D. Wilmaerts, E. Louwagie, P. Herpels, P. and Matthay, L. Khodaparast, F. Rousseau, J. Schymkowitz, and J. Michiels. The dynamic transition of persistence towards the vbnc state during stationary phase is driven by protein aggregation. *bioRxiv*, 2021.
 - ¹⁴ Charlène Planchenault, Marine C Pons, Caroline Schiavon, Patricia Signier, Jérôme Rech, Catherine Guynet, Julie Dauverd-Girault, Jean Cury, Eduardo PC Rocha, Ivan Junier, et al. Intracellular positioning systems limit the entropic eviction of secondary replicons toward the nucleoid edges in bacterial cells. *Journal of Molecular Biology*, 2020.

Reviewers' Comments:

Reviewer #2:

Remarks to the Author:

The authors have adequately addressed the issues noted in my previous report.
I recommend publication of the manuscript in Nat. Commun.

Reviewer #3:

Remarks to the Author:

I would support the publication of this manuscript if the authors will add complexity to their system as they have indicated in their rebuttal letter:

"(a) Introducing small inert molecules, like dextran, to simulate the effect of molecular crowding and
(b) exploring demixing in systems that involve multiple plasmids with fixed and variable size."

Moreover, it is true that protein aggregation happens at higher frequencies in persister and viable but non culturable bacteria. However, it is a much broader phenomenon that can be enhanced by a variety of environmental stimuli, please see:

<https://doi.org/10.1021/acsinfecdis.1c00154>

<https://doi.org/10.1093/femsre/fuz026>

<https://royalsocietypublishing.org/doi/10.1098/rstb.2018.0442>

<https://doi.org/10.1371/journal.pbio.2003853>

<https://doi.org/10.1073/pnas.0708931105>

Therefore, I encourage the authors to add this discussion to their manuscript since I think (and the authors seem to agree) that this is very relevant to their findings and vice versa this field could benefit from their findings thus broadening the scope of their manuscript.

Response to Reviewers

We thank reviewers for their constructive comments and suggestions. In response to reviewer 3, we have chosen to increase our system complexity by adding small molecules (dextran) to simulate macromolecular crowding. The result is shown in the new subsection *Effect of Macromolecular Crowding on Plasmid Distribution*. We also present results in this reviewer reply on introducing multiple plasmids into our experiment. However, while we are of course open to the reviewer's view here, we think that the multiple plasmid results, while substantive, will make the present manuscript a little too lengthy, and are better saved for a follow-up publication.

I. REVIEWER 1

Reviewer 1 did not respond to our first rebuttal and resubmission, so as suggested by the editor we modified the manuscript according to the comments of reviewer 3 (note that we have already addressed reviewer 2's concerns). In response to reviewer 1, we will briefly reiterate the points we made at greater length in our earlier rebuttal. Overall, we feel that (1), while it is agreed that simulation based methodologies have been extensively used to study organization of multiple polymers in confinement, our study provides an aspect missed from previous investigations on this topic: a model experimental system, containing a precisely calibrated degree of complexity, which can serve as a stepping stone to modeling the full complexity of an *in vivo* biological system. (2) We believe it is surprising and significant that this simple nanofluidic model, containing just a single plasmid and T₄ in a cavity, can generate an annular and polar distribution of the plasmid resembling *in vivo* observations. This suggests that generic physical interactions (e.g. repulsive excluded volume interactions and interactions with the confining potential) may be behind the observations of plasmid segregation *in vivo*, which have attracted considerable recent attention. Finally, we note that our suggested addition to the present study to address reviewer 3's comments adds another surprising and significant feature, namely that confinement anisotropy can enhance the effect of molecular crowding.

II. REVIEWER 2

The authors have adequately addressed the issues noted in my previous report. I recommend publication of the manuscript in Nat. Commun.

Thank you!

III. REVIEWER 3

I would support the publication of this manuscript if the authors will add complexity to their system as they have indicated in their rebuttal letter:

Thank you for your positive comments. We have attempted to add complexity as indicated below:

(a) Introduce small inert molecules, like dextran, to simulate the effect of molecular crowding:

We have added a new sub-section on introducing dextran into our system (*Effect of Macromolecular Crowding on Plasmid Distribution*, text and accompanying figure shown below). We find, overall, that crowding alters the system in a manner strongly dependent on cavity anisotropy, which we believe is a significant additional finding:

We introduce small inert molecules (dextran, gyration radius ~ 2.6 nm) into the plasmid-T₄-DNA confinement system to simulate the effect of molecular crowding. We observe that a high concentration of dextran (volume fraction $v_\phi = 6.3 \cdot 10^{-2}$) alters the plasmid probability density in a manner that depends on the overall cavity anisotropy (Fig. 1a). Specifically, for circular and anisotropic cavities, crowd-ers displace the plasmid probability density inwards from the cavity edges while also enhancing segregation of plasmids from the cavity center towards the cavity edges. We quantify the observed phenomenon by measuring, for a circular cavity, a radially averaged

plasmid probability density (Fig. 1c) and, for the anisotropic elliptical cavity, the plasmid probability density along cross-sections parallel and perpendicular to the cavity major axis (Fig. 1b) as well as the probability density averaged along an elliptical contour (Fig. 1d). We plot the probability density averaged over the elliptical contours versus an effective radial coordinate defined as $r_{\text{eff}} = \sqrt{\frac{x^2}{a^2} + \frac{y^2}{b^2}}$, where a and b correspond to the length of semi-major and semi-minor axis respectively. For purposes of quantifying shifts in the plasmid probability density, the distribution edge is defined as the position where the average probability density (in Fig. 1c,d) is equal to $\frac{1}{e}$ of its maximum. The effect of crowding is small for the symmetric cavity; here the plasmid probability densities with and without dextran are qualitatively similar, with only a small enhancement at the cavity edges present (Fig. 1c). For the symmetric cavity, crowders displace the plasmid probability density inward by $0.04 \pm 0.01 \mu\text{m}$. However, the presence of anisotropy ($e = 0.9$) strongly enhances the effect of crowding. In the anisotropic cavity, the inwards displacement is a factor of four greater ($0.16 \pm 0.01 \mu\text{m}$ from the probability density averaged over elliptical cross-section, for comparison displacement along the minor axis is $0.12 \pm 0.03 \text{ nm}$ and the displacement along major axis is $0.180 \pm 0.03 \text{ nm}$). Additionally, the segregation effect, towards both the cavity periphery and poles, is amplified (Fig. 1a,d).

Neutral dextran nanoparticles are expected to influence the system purely entropically.¹⁻³ While crowders promote the compaction of T_4 -DNA, yielding more space accessible to the plasmid, the crowders can also accumulate at the cavity perimeter,² reducing the accessibility of the cavity edge. The plasmids then tend to occupy an intermediate region between the cavity edge, with its high concentration of crowders, and the central region of the cavity occupied by the T_4 -DNA. The increased overall compaction of the T_4 -DNA increases the T_4 -DNA concentration in the cavity center and thus the influence of excluded-volume, enhancing the segregation effect of the plasmids relative to the T_4 -DNA. Critically, anisotropy enhances the effect of crowding (in Fig. 1a, eccentricity= 0.9). One possible explanation is that the chain conformation itself has a transient anisotropy⁴ with increased elongation along a particular axis. In the symmetric cavity, this elongated axis can rotate freely, so that the plasmid and the crowders, which are much smaller than T_4 -DNA, can maximize their accessible volume by moving in a coordinated fashion with the T_4 -DNA (transiently occupying regions to either side of the elongated chain). However, in the anisotropic cavity, the more compact T_4 -DNA aligns with the cavity major axis and is rotationally constrained. The crowders are thus forced to accumulate preferentially towards the cavity edge.

In addition we have added the following comments to the discussion:

“Our findings additionally suggest that molecular crowding may influence the degree of plasmid segregation and polar accumulation and that the effects of crowding are enhanced by cavity anisotropy.”

and also:

“Our nanofluidic model permits a wide-range of additional experiments that can, following a “bottom-up philosophy”, explore the global significance of additional biological complexity on the overall entropy driven chain demixing. Our addition of crowding agents to the plasmid- T_4 -DNA model is an example of how we can increase system complexity step-by-step.”

(b) Explore demixing in systems that involve multiple plasmids:

We address this point by performing multi-plasmid confinement experiments, with and without T_4 DNA present, in rectangular cavities containing an end-cap geometry in the form of a circular section. We decided to focus on the case of multiple identical plasmids rather than plasmids of varying size. In the absence of T_4 -DNA, unsurprisingly, we observe that a single plasmid has a uniform distribution (Fig. 2a, b), and the distribution for two plasmids also appears to be uniform. However, if three plasmids are confined, we observe there is a slight tendency that the plasmids are found at the cavity edge with greater probability. We find this surprising. While it is well known that a single particle in a confined cavity will have a preference for the cavity edge in the presence of crowding agents that induce a depletion interaction between the particle and cavity edge⁵, no crowding agents are present here. We believe that the present effect arises from the multiple plasmids inducing depletion interactions with respect to each other. Conversely, if T_4 DNA is present, then we observe that increasing the number of plasmids makes the probability distribution more homogenous (see Fig. 2a, b). We attribute this behavior to repulsive interactions between the plasmids competing with repulsive interactions arising between the plasmids and the T_4 , tending to push the plasmids farther out from the cavity edge into regions where a higher T_4 DNA concentration is present.

The presence of multiple plasmids also affects the system dynamics. Figure 2b presents results on the dwell time

FIG. 1: **a.** Plasmid probability density in cavity, obtained from histogrammed plasmid position measurements, with and without crowding and for a symmetric and anisotropic cavity. The crowder volume fraction is $v_\phi = 6.3 \cdot 10^{-2}$ and the scale bars are $1 \mu\text{m}$. The red dashed line gives the cavity edge. **b.** Cross-section of the plasmid position probability along the cavity major axis (upper panel) and along the cavity minor axis (lower panel). **c.** Radially averaged probability density of plasmid position in the absence and presence of crowding. The inset shows the contour along which the probability density average is taken. **d.** Plasmid probability density averaged over an elliptical contour versus effective radial coordinate $r_{\text{eff}} = \sqrt{\frac{x^2}{a^2} + \frac{y^2}{b^2}}$, in the absence and presence of crowding (a and b correspond to the length of semi-major and semi-minor axis respectively). The inset shows the elliptical contours along which the probability density average is taken.

for two plasmids remaining in the pole in the presence of T_4 -DNA. Specifically, the two plasmid dwell time is the time over which both plasmids are present at the pole region (ending when any one of the two plasmids escapes the pole). The presence of the additional plasmid reduces the dwell-time relative to what is expected from a single plasmid: the average two-plasmid dwell time is $\langle \tau_2 \rangle = 0.18 \pm 0.01 \text{ s}$ while the average single-plasmid dwell time is $\langle \tau_1 \rangle = 0.6 \pm 0.01 \text{ s}$. Some of this difference is related to a trivial fact that statistically we expect the dwell-time of two uncorrelated plasmids to be lower by a factor of 2 (simply because both have an opportunity to escape). However, note that $\langle \tau_1 \rangle / 2 = 0.3 \pm 0.01 \text{ s}$, which is still higher than the measured $\langle \tau_2 \rangle$; this suggests that interactions are present between the two plasmids that tends to destabilize the free energy well and make escape more probable.

We believe that the above measurements demonstrate that our system can be extended to more complex systems; however, we think that the results are better saved for a follow-up paper, given natural manuscript length limitations.

FIG. 2: **a**. Plasmid probability density, in the presence/absence of T₄-DNA, for a rectangular cavity with an end-cap geometry in the form of a circular cap; a variable number of plasmids are present. The radius of the end-caps $r = 0.5 \mu\text{m}$. The red dashed line indicates the cavity wall. The scale bars are $1 \mu\text{m}$. **b**. The cross-section of the plasmid probability density taken along an axis transverse to the cavity long-axis. Each cross-section is normalized to the value of the probability density at the cavity center. **c**. Dwell time probability distribution for two plasmids residing at the poles for cavities of the same geometry as shown in (a). Dashed-lines give an exponential fit.

Moreover, it is true that protein aggregation happens at higher frequencies in persister and viable but non culturable bacteria. However, it is a much broader phenomenon that can be enhanced by a variety of environmental stimuli, please see:

- <https://doi.org/10.1021/acsinfectdis.1c00154>
- <https://doi.org/10.1093/femsre/fuz026>
- <https://royalsocietypublishing.org/doi/10.1098/rstb.2018.0442>
- <https://doi.org/10.1371/journal.pbio.2003853>
- <https://doi.org/10.1073/pnas.0708931105>

Therefore, I encourage the authors to add this discussion to their manuscript since I think (and the authors seem to agree) that this is very relevant to their findings and vice versa this field could benefit from their findings thus broadening the scope of their manuscript.

We agree with the reviewer on this point and have added an expanded discussion of the protein aggregation phenomena, and its relevance to our technology, to our discussion:

“A second potential application of our system is to attempt to elucidate certain poorly understood *in vivo* phenomena that appear impacted by nucleoid exclusion. In particular, the formation of localized aggregates of unfolded/misfolded protein is a widespread phenomena in bacteria.⁶ These aggregates often appear at the cell-poles (as in the case in *E. coli*⁶), and form in response to proteotoxic stresses arising from cellular and environmental factors, for example cell aging,⁷ decline in ATP levels,⁶ heat shock,⁸ antibiotic treatment and high levels of heterologous protein expression.⁹ The aggregates are inheritable and associated with increased resistance to stress⁸ with a close connection to persister phenotypes that can survive starvation and exposure to high levels of antibiotics.^{9–11} The aggregates have been observed to freely diffuse in nucleoid free regions of the bacteria,^{12,13} suggesting that entropic forces may play a role in their polar localization, analogous to the localization of the plasmids. In principle, protein aggregates, for example extracted from bacteria via centrifugation¹⁴ and labeled via IbpA-YFP fusion proteins,⁸ could be introduced to our nanofluidic system to observe if polar organization occurs in the absence of any active mechanisms. Protein aggregates in non-stressed conditions often form at only *one* of the cell poles.⁷ We suggest this may result due to competition between protein aggregates and other macromolecular components, such as plasmids, for polar locations (a hypothesis our nanofluidic model allows us to partially test, by exploring systems containing mixtures of plasmids and protein aggregates).”

-
- ¹ J. Kim, C. Jeon, H. Jeong, Y. Jung, and B.-Y. Ha, *Soft Matter* **11**, 1877 (2015).
² C. Zhang, P. G. Shao, J. A. van Kan, and J. R. van der Maarel, *Proceedings of the National Academy of Sciences* **106**, 16651 (2009).
³ J. Pelletier, K. Halvorsen, B.-Y. Ha, R. Paparcone, S. J. Sandler, C. L. Woldringh, W. P. Wong, and S. Jun, *Trends in Microbiology* **109**, E2649 (2012).
⁴ D. J. Bonthuis, C. Meyer, D. Stein, and C. Dekker, *Phys. Rev. Lett.* **101**, 108303 (2008).
⁵ A. Dinsmore, D. Wong, P. Nelson, and A. Yodh, *Physical Review Letters* **80**, 409 (1998).
⁶ F. D. Schramm, K. Schroeder, and K. Jonas, *FEMS Microbio. Rev.* **44**, 54 (2020).
⁷ A. B. Lindner, R. Madden, A. Demarez, E. J. Stewart, and F. Taddei, *Proceedings of the National Academy of Sciences* **105**, 3076 (2008).
⁸ S. K. Govers, J. Mortier, A. Adam, and A. Aertsen, *PLoS biology* **16**, e2003853 (2018).
⁹ O. Goode, A. Smith, U. Lapińska, R. Bamford, Z. Kahveci, G. Glover, E. Attrill, A. Carr, J. Metz, and S. Pagliara, *ACS Infectious Diseases* **7**, 1848 (2021).
¹⁰ Y. Pu, J. X. Li, Y. and, T. Tian, Q. Ma, Z. Zhao, S.-Y. Lin, Z. Chen, B. Li, G. Yao, M. C. Leake, et al., *Molecular Cell* **73**, 143 (2019).
¹¹ L. Dewachter, C. Bollen, D. Wilmaerts, E. Louwagie, P. Herpels, P. and Matthay, L. Khodaparast, F. Rousseau, J. Schymkowitz, and J. Michiels, *mBio* **12**, e00703 (2021).
¹² A.-S. Coquel, J.-P. Jacob, M. Primet, A. Demarez, M. Dimiccoli, T. Julou, L. Moisan, A. B. Lindner, and H. Berry, *PLoS computational biology* **9**, e1003038 (2013).
¹³ A. Gupta, J. Lloyd-Price, R. Neeli-Venkata, S. M. Oliveira, and A. S. Ribeiro, *Biophysical journal* **106**, 1928 (2014).
¹⁴ E. Maisonneuve, L. Fraysse, D. Moinier, and S. Dukan, *J. of Bacteriology* **190**, 887 (2008).

Reviewers' Comments:

Reviewer #3:

Remarks to the Author:

The authors have convincingly addressed issues from the previous round of review. I am happy to recommend the manuscript for publication provided that the authors address the issues below.

The abstract would benefit from a concluding sentence about the broader implications of this study.

"As prokaryotes lack membrane compartments, these multiple dsDNA molecules are free to interact." This sentence is incorrect GN bacteria for example display multiple membranes.

"...obscure understanding of the larger phase-space." Could the authors explain more clearly what do they mean

"Exploring the physical mechanism over a larger parameter space|some regions of which may have no clear biological relevance|is essential to probe the underlying physics and can place existing in vivo systems in a larger context, for example shedding light on differences between species that occupy different points in parameter space, or physical constraints critical for cellular viability.". This paragraph is very important but not accessible by the general audience as it is currently written

Line 118, "a phenomena" should be "a phenomenon"

"Once driven beneath the flexible membrane via pressure actuated flow, the molecules are isolated in the cavities by depressing the lid, a procedure that is repeated until two differentially stained molecules are trapped, enabling independent monitoring of their conformation." Could the authors add a statement on how they make sure this process does not pose a mechanical stress on the molecules

"a sharp peak (or dip)" Could the authors please clarify what determines whether a peak or a dip appear

"we explore the interaction between a larger, linear DNA molecule (T4-DNA, 166 kbp) and a plasmid vector (pBR322, 4361bp) confined in an elliptical cavity" could the authors clarify if these molecules are charged or what other types of interactions the authors expect.

"a repulsive interaction of the plasmid with the cavity boundary". I am assuming the authors are referring to an repulsive electrostatic interaction? As suggested above the authors should clarify the charge of each molecule and surface investigated and make predictions on the corresponding interactions.

The discussion could be expanded to cover predictions for bacteria with different shapes. Indeed the authors explore slits with different eccentricities. The authors could use their data to make predictions on rod-shaped vs sphere-like bacteria, i.e. *E. coli* vs *S. aureus*? Also could the authors comment on whether a double vs single bacterial membrane would have an impact on their predictions.

Line 674: *E. coli* should be reported in italic

"for example cell aging,⁵⁴" the authors should acknowledge that cell ageing does not always cause the formation of aggregates <http://dx.doi.org/10.1098/rstb.2018.0442> this correlation is indeed under debate and the authors platform could play an important role in clarifying this issue.

Response to Reviewers

We thank reviewers for their constructive comments and suggestions.

I. REVIEWER 3

The authors have convincingly addressed issues from the previous round of review. I am happy to recommend the manuscript for publication provided that the authors address the issues below.

Thank you!

The abstract would benefit from a concluding sentence about the broader implications of this study.

We have modified the concluding sentence of the abstract, indicating the connection to *in vivo* observations (added text in italics): “We find that the plasmid is excluded from the larger molecule and will exhibit a preference for the ellipse poles, *giving rise to a non-uniform spatial distribution in the cavity that may help explain the non-uniform plasmid distribution observed during in vivo imaging of high-copy number plasmids in bacteria.*”

“As prokaryotes lack membrane compartments, these multiple dsDNA molecules are free to interact.” This sentence is incorrect GN bacteria for example display multiple membranes.

We are simply referring to the fact that bacteria lack a separate nuclear compartment. We have rewritten the sentence as follows: “As prokaryotes lack a separate nuclear compartment, these multiple dsDNA molecules are free to interact within the cell volume.”

“...obscure understanding of the larger phase-space.” Could the authors explain more clearly what do they mean

“Exploring the physical mechanism over a larger parameter space—some regions of which may have no clear biological relevance—is essential to probe the underlying physics and can place existing in vivo systems in a larger context, for example shedding light on differences between species that occupy different points in parameter space, or physical constraints critical for cellular viability.”. This paragraph is very important but not accessible by the general audience as it is currently written

We are addressing these two comments together as they are closely linked. We think the issue is that the language “phase-space” is physics jargon and confusing to a more general audience. Thus we have replaced the one instance of phase-space by parameter space. We also feel the concept of parameter space needs to be more concretely defined. In particular, by “parameter space,” we mean that we are considering a cell as a system defined by a series of gross biophysical parameters such as cell size/degree of anisotropy, number of chromosomes/plasmids, sizes of chromosomes/plasmids, plus crowding concentration etc., which we might imagine as living in a very complicated (i.e. very high dimensional) parameter space. Of course, biological systems occupy only very small regions of this space, because the biological systems are only viable in a narrowly defined region of the space. Our point is that it is in fact interesting to consider the physical behavior over larger regions of parameter-space, even parts of the space that do not correspond to viable organisms, because this might give us insight into why the *in vivo* systems occupy the portions of the parameter space they do.

We have changed the text as follows:

Lastly, focusing only on specific *in vivo* systems may obscure understanding of how the system behaves physically over a larger parameter space (i.e. a parameter space defined in terms of gross biophysical parameters like cell size, degree of anisotropy in the cell geometry, number of chromosomes/plasmids, sizes of chromosomes/plasmids and degree of crowding). Specific *in vivo* systems occupy only narrowly defined regions of this space. However, exploring the physical behavior over much larger portions of the space, even parts of the space that do not contain viable organisms, is essential to probe the underlying physics and can place existing *in vivo* systems in a larger context, for example shedding light on differences between species that occupy different points in parameter space,¹ or physical constraints critical for cellular viability.

Line 118, “a phenomena” should be “a phenomenon”

We have corrected this typo. Thank you!

"Once driven beneath the flexible membrane via pressure actuated flow, the molecules are isolated in the cavities by depressing the lid, a procedure that is repeated until two differentially stained molecules are trapped, enabling independent monitoring of their conformation." Could the authors add a statement on how they make sure this process does not pose a mechanical stress on the molecules.

If the process of trapping induced damage, we would expect to see molecule fragmentation immediately when we release the chains from the cavity after trapping. We confirm that the chains are intact, i.e. not broken, once the membrane is depressed and the chains are confined in the cavity. In fact, we have never observed a molecule fragmenting after trapping.

"a sharp peak (or dip)" Could the authors please clarify what determines whether a peak or a dip appear

Whether we see a peak/dip depends on how we define the separation vector between the chain center-of-mass positions. The projection of this vector along the cavity axis is positive when the YOYO-1 stained chain (green) is top and the YOYO-3 stained chain (red) is bottom, and negative for the opposite situation (see Fig. 3 in the manuscript). When the separation vector has an initially positive projection (green top, red bottom), and the molecules attempt to flip but fail, this creates a short dip in the projection. When the separation vector has an initially negative projection (green bottom, red top), and the molecules attempt to flip but fail, this creates a short peak in the projection.

"we explore the interaction between a larger, linear DNA molecule (T₄-DNA, 166 kbp) and a plasmid vector (pBR322, 4361bp) concerned in an elliptical cavity" could the authors clarify if these molecules are charged or what other types of interactions the authors expect.

The DNA samples are negatively charged under our buffer condition (1X Tris buffer. pH=8.0). Under these conditions, in the absence of crowding agents, the dsDNA molecules (plasmid+T₄) interact internally and with each other purely through short-ranged electrostatic repulsion. This gives rise to enhanced excluded-volume interactions arising from the entropy loss due to regions around the chain that are excluded to other parts of the same or different chains. In particular, the interactions are short-ranged as the electrostatics are screened over the Debye length, which determines how far an electric field can extend in salt solution. The fact that the electrostatic interactions are short-range means that they can be accurately quantified by assigning the dsDNA molecules an effective width w_{eff} ,² which includes the intrinsic physical width of the molecule (2 nm) plus an effective electrostatic interaction range (in our buffer conditions the effective width is around 10 nm).³ We note that there are classic biophysical models for the effective width that quantify its dependence on DNA charge and Debye length.⁴ The effective width then leads to the chain exhibiting a higher excluded volume incorporating the effect of electrostatic repulsion. In particular, the radius of gyration of a self-avoiding polymer coil scales as $\sim w_{\text{eff}}^{\frac{1}{2}}$,² which we have incorporated in our estimate of the plasmid gyration radius (lines 362-365). The plasmid gyration radius then determines the volume excluded by the plasmid relative to the larger DNA coil, which is the key parameter determining the interaction potential of the plasmid in the presence of the T₄-DNA molecule.

"a repulsive interaction of the plasmid with the cavity boundary". I am assuming the authors are referring to a repulsive electrostatic interaction? As suggested above the authors should clarify the charge of each molecule and surface investigated and make predictions on the corresponding interactions.

We expect that the boundary DNA interactions involve primarily short-ranged repulsive electrostatic interactions between the wall and the DNA molecule and more subtly the degree to which the plasmid can be compressed as it is squeezed against the cavity wall. From the point of view of the current study, the main question is what determines the length scale of the interaction between the plasmid and the wall, and this length scale is likely on order of the plasmid gyration radius, because the plasmid gyration determines how close the plasmid center position can approach the wall. While short-range electrostatic repulsion will increase this estimation slightly, we note that the wall-DNA interaction range is on order of the effective width, and the effective width ($\sim 10\text{ nm}$) represents less than a 10% correction given the value of the plasmid gyration radius. We indeed find that our fitted DNA-wall interaction range is on order of the plasmid gyration radius (see discussion lines 347-368). We feel that a more detailed theoretical investigation of the wall-plasmid interaction is beyond the scope of the present study and is well suited to simulation, which can then be bench-marked against our results. A simulation study can incorporate subtle

effects such as the degree to which the plasmid coil is squeezed as it approaches the wall. In particular, we note that our key conclusions are not dependent on the details of the wall-DNA interactions, as we get similar results for interactions of exponential and WCA (Weeks-Chandler-Andersen) form.

The discussion could be expanded to cover predictions for bacteria with different shapes. Indeed the authors explore slits with different eccentricities. The authors could use their data to make predictions on rod-shaped vs sphere-like bacteria, i.e. E. coli vs S. aureus? Also could the authors comment on whether a double vs single bacterial membrane would have an impact on their predictions.

S. aureus is known to divide in three successive orthogonal planes, each plane orthogonal to that of the previous generation. Certainly, given the spherical geometry, it is difficult to see a role for entropy in this case, and one would be inclined to argue that the mechanism must be purely active. However, a recent study of the related cocci *D. radiodurans* suggests that the situation is not so straightforward. In particular, these authors separately imaged the nucleoid structures within the membrane during division⁵. The membranes are in fact not perfectly hemispherical during division. Starting out approximately hemispherical in division phase 1 (symmetric diad), the individual cells become quasi ellipsoidal at the point at which the new septa are apparent. Remarkably, the nucleoids are often elongated at this point. This suggests, in this particular case, that transient anisotropy in cell geometry during division is associated with nucleoid partitioning, even in a bacteria with a spherical geometry during stationary phase, which is consistent with what we would predict is necessary if entropic effects play some role. Regarding plasmid partitioning, it seems likely that the transient anisotropy might also lead to the plasmids adopting a polar preference during the division process.

We do not feel that the question of a double vs single bacterial membrane should impact our conclusions as they depend only on the available cell volume, not the exact nature of the confining membrane.

Line 674: E. coli should be reported in italic

We have corrected the typo.

"for example cell aging" the authors should acknowledge that cell aging does not always cause the formation of aggregates <http://dx.doi.org/10.1098/rstb.2018.0442> this correlation is indeed under debate and the authors platform could play an important role in clarifying this issue.

Following the reviewer's suggestion, we have rewritten the sentence in the following way to take into account the uncertainties over the correlation between aggregate formation and cell aging: "These aggregates often appear at the cell-poles (as in the case in *E. coli*⁶), and form in response to proteotoxic stresses arising from cellular and environmental factors, for example decline in ATP levels,⁶ heat shock,⁷ antibiotic treatment, high levels of heterologous protein expression⁸ and potentially cell aging⁹ (although the correlation of aggregate formation with cell aging is under debate¹⁰)."

¹ S. Jun and A. Wright, Nature Reviews Microbiology **8**, 600 (2010).

² W. Reisner, J. N. Pedersen, and R. H. Austin, Reports on Progress in Physics **75**, 106601 (2012).

³ A. R. Klotz, L. Duong, M. Mamaev, H. W. de Haan, J. Z. Chen, and W. W. Reisner, Macromolecules **48**, 5028 (2015).

⁴ D. Stigter, Biopolymers: Original Research on Biomolecules **16**, 1435 (1977).

⁵ K. Floc'h, F. Lacroix, P. Servant, Y.-S. Wong, J.-P. Kleman, D. Bourgeois, and J. Timmins, Nature communications **10**, 1 (2019).

⁶ F. D. Schramm, K. Schroeder, and K. Jonas, FEMS Microbio. Rev. **44**, 54 (2020).

⁷ S. K. Govers, J. Mortier, A. Adam, and A. Aertsen, PLoS biology **16**, e2003853 (2018).

⁸ O. Goode, A. Smith, U. Łapińska, R. Bamford, Z. Kahveci, G. Glover, E. Attrill, A. Carr, J. Metz, and S. Pagliara, ACS Infectious Diseases **7**, 1848 (2021).

⁹ A. B. Lindner, R. Madden, A. Demarez, E. J. Stewart, and F. Taddei, Proceedings of the National Academy of Sciences **105**, 3076 (2008).

¹⁰ U. Łapińska, G. Glover, P. Capilla-Lasheras, A. J. Young, and S. Pagliara, Philosophical Transactions of the Royal Society B **374**, 20180442 (2019).